# DiffBED: Scaling Bayesian Experimental Design to High-Dimensions

**Adhithya Saravanan**[*]    **Rik Knowles**[*]    **Gavin Kerrigan**    **Tom Rainforth**
Department of Statistics
University of Oxford
`{saravanan,knowles,kerrigan,rainforth}@stats.ox.ac.uk`

## Abstract

Bayesian experimental design (BED) is a principled framework for intelligent data acquisition. However, current approaches do not scale to problems with high–dimensional designs, impeding its uptake. We show that this limitation arises predominantly from the difficulty in specifying a likelihood model that remains accurate throughout the design space, and that without this, standard design optimisation procedures lead to a reward-hacking-like behaviour that exploits deficiencies in the likelihood, producing implausible or unrealistic designs. To overcome this, we introduce `DiffBED`, an approach based on a novel BED objective that explicitly rewards realistic designs. Realism is captured by a diffusion model, which we guide using information-theoretic experimental design criteria to generate highly informative yet realistic designs. This enables BED at an unprecedented scale: while existing applications of BED have been restricted to design spaces with a handful of dimensions, we show that `DiffBED` can successfully scale to designing high–resolution images.

## 1 Introduction

Experimentation, the process by which we gather information about a phenomenon of interest, is a central task throughout science and industry. In scenarios where data collection is costly or time-consuming, such as drug discovery (Paul et al., 2010; DiMasi et al., 2016) or clinical trials (Fogel, 2018), it is natural to seek designs that yield data that is *maximally informative*. This intuition is captured by the framework of Bayesian experimental design (BED) (Lindley, 1956; Chaloner & Verdinelli, 1995; Rainforth et al., 2024; Huan et al., 2024). In BED, we specify a probabilistic model of the data gathering process, use this to derive a formal notion of the *expected information gain* (EIG) of an experiment for a target quantity of interest, then optimise this EIG to yield designs we expect to maximally reduce our uncertainty. Thanks to the coherence of Bayesian reasoning, this framework is naturally suited to adaptively gathering information across several experimental steps, utilising information from previous experiments in each sequential decision we make.

Although, in principle, BED can be applied to a wide array of tasks, successful applications have historically been limited to simple problems in which the design variables are low-dimensional (Papadimitriou, 2004; Loredo, 2004; Vanlier et al., 2012; Shababo et al., 2013; Myung et al., 2013; Vincent & Rainforth, 2017; Watson, 2017; Dushenko et al., 2020). Developing methods for high-dimensional design spaces is thus a critical open challenge (Rainforth et al., 2024; Huan et al., 2024), with the problem historically being considered mostly as one of developing scalable EIG estimators (Foster et al., 2019; Goda et al., 2022; Ao & Li, 2024; Huan et al., 2024; Iollo et al., 2025a).

In this work, we identify an even more fundamental barrier: the difficulty of specifying a likelihood that faithfully reflects the real-world data-generating process across the entire design space. As the dimensionality of the design increases, model misspecification becomes increasingly unavoidable, since the likelihood must remain accurate over an exponentially growing space. While modern machine learning models succeed by learning structure near the data manifold, experimental design explicitly seeks out information that goes beyond what is already known. This inevitably requires the likelihood to extrapolate into regions where our modelling assumptions are most fragile.

---

[*]Equal contribution.

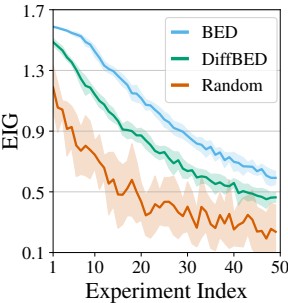 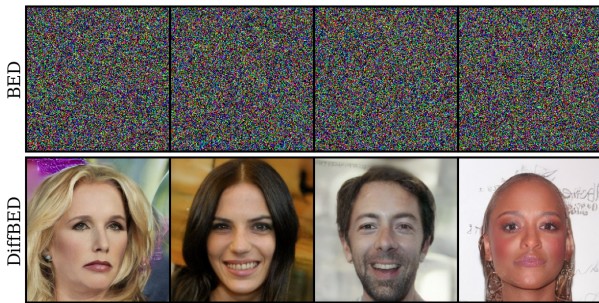

Figure 1: The incremental EIG achieved at each experiment iteration (left) and initial design set chosen (right) for the *search* experiment presented in Section 6.2, where an eyewitness is asked to provide preferences between a set of generated images based on their likeness to a suspect. We see that, despite having high EIG values, the designs chosen by BED are meaningless pixel noise, but those provided by our `DiffBED` approach are still realistic and helpful images.

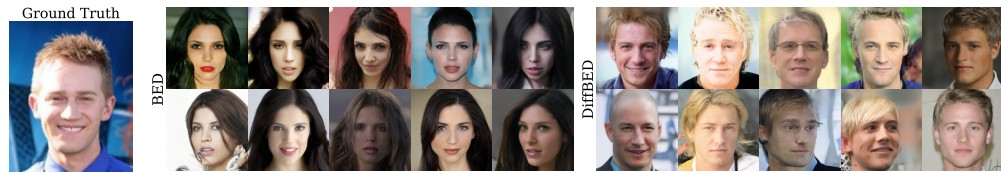

Figure 2: Posterior samples after 50 search iterations for standard BED (left) and `DiffBED` (right).

The consequence of this previously undocumented issue, illustrated in Figure 1, is that directly optimising the EIG causes imperfections in the likelihood to be exploited, resulting in meaningless designs. This failure mode is analogous to reward hacking in reinforcement learning (Manheim & Garrabrant, 2019; Skalse et al., 2022). Namely, although existing stochastic gradient methods are fully capable of optimising the EIG even in very high-dimensional settings, the resulting designs are unrealistic and lie in regions where the likelihood is spuriously overconfident, with the corresponding experimental outcomes being, in reality, uninformative. When BED is applied sequentially to adaptively select designs, this issue is further amplified. As shown in Figure 2, the resulting designs induce a pathological feedback loop in posterior updates, where rather than remaining uncertain, the posterior collapses around an *incorrect* value.

To overcome these challenges, we introduce `DiffBED`, a novel method for BED in high-dimensional design spaces. While the assumed likelihood is typically misaligned globally, we argue that the likelihood is often sufficiently well-specified on a manifold of realistic and feasible designs. As such, `DiffBED` revisits the fundamentals of BED by proposing a new utility function that explicitly rewards designs which are both highly informative and realistic, and hence, lie in regions of low misalignment. In many practical settings, we have prior knowledge about which designs are feasible, for example through access to unlabelled data or a foundational generative model over the design space, which allows this realism criterion to be specified independently of the experimental outcomes.

`DiffBED` uses a diffusion model for its design prior. Designs are then generated by a process we call *information-guided diffusion*, where designs are chosen by simulating the reverse–time SDE of this diffusion process, with guidance provided by an estimator for the score of the EIG. This estimator is itself based on a combination of Tweedie's formula (Robbins, 1956) and existing EIG gradient estimators (Rainforth et al., 2018; Foster et al., 2020). Adaptive design then simply updates the EIG estimator to incorporate new observations and reruns the diffusion with this new guidance.

As shown in Figure 1, `DiffBED` produces designs that are meaningful and informative, which in turn enables effective learning about the target quantity of interest (see Figure 2). `DiffBED` therefore represents the first successful application of BED in high-dimensional design spaces: we show successful deployment of `DiffBED` to design spaces in excess of $750\,000$ dimensions, whereas previous BED approaches have rarely been successfully used beyond $\sim 20$ dimensions. More broadly, this work shows that coupling BED with powerful generative models provides a practical pathway for scaling principled experimental design to realistic, high-dimensional problems, substantially expanding the scope and applicability of BED.

## 2 PRELIMINARIES

We begin by reviewing the key concepts underpinning the BED framework. We express our initial beliefs about the target variable of interest, $\theta$, through a prior distribution $p(\theta)$. We also specify a likelihood $p(y \mid \theta, \xi)$, which gives the probability of possible experimental outcomes $y$ given $\theta$ and a design $\xi$. If an outcome $y$ were observed by running an experiment with design $\xi$, the *information gain* (IG) of such an experiment is the reduction in entropy obtained when updating from the prior $p(\theta)$ to the posterior $p(\theta \mid y, \xi)$ (Lindley, 1956), defined as, $\text{IG}(y, \xi) = \text{H}[p(\theta)] - \text{H}[p(\theta \mid y, \xi)]$, where $H[\cdot]$ denotes the Shannon entropy. Since the outcome $y$ is unknown before running the experiment, we instead maximise the *expected information gain* (EIG) (Lindley, 1972; Bernardo, 1979; Sebastiani & Wynn, 2000):

$$\text{EIG}(\xi) = \mathbb{E}_{p(y \mid \xi)}[\text{IG}(y, \xi)] = \mathbb{E}_{p(\theta)\,p(y\mid\theta,\xi)}\left[\log p(\theta \mid y, \xi) - \log p(\theta)\right], \quad (1)$$

$$= \mathbb{E}_{p(\theta)\,p(y\mid\theta,\xi)}\left[\log p(y \mid \theta, \xi) - \log p(y \mid \xi)\right], \quad (2)$$

where the last equality follows from Bayes' rule. The EIG is then maximized to produce an optimal design $\xi^* = \operatorname{argmax}_{\xi \in \Xi} \text{EIG}(\xi)$, where $\Xi$ is the space of admissible designs.

**Adaptive Design** In many applications, we are interested in producing a sequence of designs $\xi^{1:K} := (\xi^1, \xi^2, \ldots, \xi^K)$ yielding data $y^{1:K} := (y^1, y^2, \ldots, y^K)$. While the sequence $\xi^{1:K}$ could be determined statically before observing any data, a more performant approach is to select the design $\xi^k$ adaptively depending on the history $\mathcal{D}^{k-1} = \{(\xi^i, y^i)\}_{i=1}^{k-1}$ of designs and outcomes prior to step $k$. In this adaptive setting, the design for the $k$-th experiment is typically obtained by greedily maximising the *incremental* $\text{EIG}(\xi^k \mid \mathcal{D}^{k-1}) = \mathbb{E}_{p(\theta \mid \mathcal{D}^{k-1})\,p(y^k\mid\theta,\xi^k)}\left[\log p(y^k \mid \theta, \xi^k) - \log p(y^k \mid \xi^k)\right]$. This expression is equivalent to the EIG except that the prior, $p(\theta)$, is replaced with the posterior, $p(\theta \mid \mathcal{D}^{k-1})$, which reflects the current beliefs given the history (Chernoff, 1958; Chaloner & Verdinelli, 1995).

**Estimating the EIG** While the EIG is conceptually appealing, estimating it can be challenging due to the doubly intractable nature of the objective, with a wide variety of approaches proposed to address this, see (Rainforth et al., 2024, Section 3) for a review. However, when the outcome space $\mathcal{Y}$ is discrete, the outer expectation with respect to the likelihood can be enumerated over. In this case, the EIG is now singly-intractable, yielding a non-nested estimator (Rainforth, 2017; Gal et al., 2017)

$$\widehat{\text{EIG}}(\xi) = -\sum_{y \in \mathcal{Y}} \hat{p}(y \mid \xi) \log \hat{p}(y \mid \xi) + \frac{1}{N} \sum_{n=1}^{N} \sum_{y \in \mathcal{Y}} p(y \mid \theta_n, \xi) \log p(y \mid \theta_n, \xi), \quad (3)$$

where $\theta_n \sim p(\theta)$ (or $\theta_n \sim p(\theta \mid \mathcal{D}^{(k-1)})$ in adaptive settings) and $\hat{p}(y \mid \xi) = \frac{1}{N} \sum_{n=1}^{N} p(y \mid \theta_n, \xi)$. In practice, we not only want to estimate the EIG, but optimize it. Various stochastic gradient-based methods have been adapted for allowing such EIG maximisation to be done in an effective and scalable manner (Huan & Marzouk, 2014; Foster et al., 2020; Goda et al., 2022).

**Diffusion Models** are a powerful class of generative models (Sohl-Dickstein et al., 2015; Lai et al., 2025) that our approach will build on when producing designs. Diffusion models define a forward stochastic differential equation (SDE) which gradually corrupts data with noise and learn a reverse process which undoes this corruption (Ho et al., 2020; Song et al., 2021). The forward SDE is

$$\mathrm{d}x_t = f(x_t, t)\,\mathrm{d}t + g(t)\,\mathrm{d}W_t \quad x_0 \sim p_0(x_0),\ t \in [0, T], \quad (4)$$

where $p_0(x_0)$ is a data distribution, $f(x_t, t)$ is a drift vector field, $g(t)$ is a noise schedule, and $dW_t$ is an increment of a standard Wiener process. The functions $f, g$ are chosen so that $p_T(x_T)$ is approximately $\mathcal{N}(0, I)$. A generative model is obtained by solving the time-reversal of Equation (4), given by

$$\mathrm{d}x_t = \left[f(x_t, t) - g(t)^2 \nabla \log p_t(x_t)\right]\mathrm{d}t + g(t)\,\mathrm{d}\overleftarrow{W}_t, \quad (5)$$

where $x_T \sim p_T(x_T)$ and $\mathrm{d}\overleftarrow{W}_t$ is a time-reversed Brownian increment. The intractable score $\nabla \log p_t(x_t)$ is approximated by a neural network $s_\phi(x_t, t)$ with parameters $\phi$ trained via denoising score matching (Hyvärinen & Dayan, 2005; Song et al., 2021). Sampling $x_T$ from a Gaussian and integrating Equation (5) backwards in time yields samples approximately drawn from $p_0(x_0)$.

## 3 DIRECTLY OPTIMISING EIG SEEKS OUT MODEL MISSPECIFICATION

Bayesian experimental design is inherently model-based, with the EIG relying on the assumed likelihood model $p(y \mid \theta, \xi)$ and a subjective prior $p(\theta)$. As such, how well the EIG reflects true

expected utility of gathering new data will depend on the accuracy of this model, in particular how well the assumed likelihood approximates the true conditional data-generating process $p_{\text{true}}(y \mid \theta, \xi)$. While the prior provides some protection against needing the likelihood to be accurate across all $\theta$, optimising over $\xi$ requires the likelihood to remain accurate across the entire design space.

When the designs $\xi$ are high-dimensional, faithfully modelling $y \mid \theta, \xi$ becomes especially challenging and it is usually not realistic to construct a likelihood that is accurate across all $(\theta, \xi)$ pairs. Indeed, the assumed likelihood $p(y|\theta, \xi)$ in high-dimensional problems will often itself be a *learned* function derived from a pre-trained machine learning model. For example, in Figure 1 our likelihood utilises a fixed encoder that captures semantic content. In such cases the likelihood will only reflect the true data-generating mechanism in data regions near where the feature extracting component was trained.

We now show that direct optimisation of the EIG is *inherently prone* to seeking out areas of the design space where the model is misspecified, specifically, it is drawn to regions where the likelihood is *overconfident*. To do this we consider the difference in using the EIG with our model's likelihood compared with a "true" EIG that uses the unknown true underlying data distribution,

$$\text{TEIG}(\xi) := \mathbb{E}_{p(\theta)p_{\text{true}}(y|\theta,\xi)} \left[ \log p_{\text{true}}(y \mid \theta, \xi) - \log p_{\text{true}}(y \mid \xi) \right], \tag{6}$$

$$= -\mathbb{E}_{p(\theta)}[\text{H}[p_{\text{true}}(y \mid \theta, \xi)]] + \text{H}[p_{\text{true}}(y \mid \xi)], \tag{7}$$

where $p_{\text{true}}(y \mid \xi) = \mathbb{E}_{p(\theta)}[p_{\text{true}}(y \mid \theta, \xi)]$. Equation (7) and the analogous term for $\text{EIG}(\xi)$ yields

$$\text{EIG}(\xi) = \text{TEIG}(\xi) + \underbrace{\mathbb{E}_{p(\theta)} \left[ \text{H}[p_{\text{true}}(y \mid \theta, \xi)] - \text{H}[p(y \mid \theta, \xi)] \right]}_{=: \mathcal{M}(\xi)} + \text{H}[p(y \mid \xi)] - \text{H}[p_{\text{true}}(y \mid \xi)]. \tag{8}$$

This decomposition provides helpful insight into how the EIG behaves when used with an approximate model likelihood. Namely, we can view $\mathcal{M}(\xi)$ as a measure on the average degree of model overconfidence across possible $\theta$: it is zero if the likelihood matches the true data generating process and it grows as the likelihood becomes more certain than it should be. Critically, $\mathcal{M}(\xi)$ varies across designs, and its presence in the decomposition encourages designs found by directly optimizing the EIG to lie where the likelihood is overconfident.

Moreover, the remaining $\text{H}[p(y \mid \xi)] - \text{H}[p_{\text{true}}(y \mid \xi)]$ term typically provides little protection against this desire to move to regions of overconfident likelihoods. In particular, these marginal data distributions will inherently be more diffuse than the corresponding likelihoods, with the averaging over $\theta$ providing regularisation on their predictions. Thus, in high-dimensional spaces it will usually be easy to find designs where we are overconfident in $y|\theta, \xi$, but our uncertainty over $\theta$ ensures that $\text{H}[p(y \mid \xi)]$ remains high. Thus, even when the likelihood is heavily misspecified, $p(y \mid \xi)$ and $p_{\text{true}}(y \mid \xi)$ will often still be similar for most $\xi$. For example, if we have a design very far away from previous designs then a misspecified (but still sensible) model will generally produce high marginal predictive uncertainty, even though it might be very confident about $y$ when $\theta$ is known. As such, the signal from this remaining term will generally not sufficiently counteract $\mathcal{M}(\xi)$ and we can expect direct EIG optimisation to seek out designs where the likelihood is overconfident. The effect of this is clearly visible in Figure 1, where direct optimisation of the EIG has led to designs for which the likelihood is very confident about the outcomes given $\theta$, but that actually correspond to meaningless images where a human responder will not be able to provide any reliable preferences.

## 4 BAYESIAN EXPERIMENTAL DESIGN VIA INFORMATION-GUIDED DIFFUSION

### 4.1 MISALIGNMENT-AWARE DESIGN SELECTION

Simply improving the fidelity of $p(y \mid \theta, \xi)$ is not a viable solution in general for overcoming such issues with misalignment: in high-dimensional design spaces, it is not usually realistic to specify a likelihood that is accurate everywhere, and optimisation can amplify even small residual errors. Instead, we accept that misalignment is unavoidable and modify the design objective itself.

Our key observation is that misalignment is not uniform across designs $\xi$; there will usually be regions where the likelihood is more reliable and less reliable, e.g. the likelihood will be more reliable for designs close to any data manifold used in its construction. As this reliability is latent, we introduce a surrogate penalty $r(\xi)$ that penalises designs for lying outside the regions where we expect good alignment. We then balance informativeness with the surrogate penalty by maximising

$$U(\xi) = \alpha \cdot \text{EIG}(\xi) - r(\xi) \tag{9}$$

where $\alpha > 0$ is hyper-parameter introduced to trade off informativeness against adherence to low misalignment regions. In complex settings, for example where designs are sets of images, a sensible closed form surrogate penalty cannot usually be directly defined. However, in such settings, we often have access to a large set of realistic and feasible designs through unlabelled auxiliary data, such as a corpus of natural images, which we can use to *learn* a surrogate penalty. For most likelihoods, especially those parametrised by learned functions, we expect the misalignment to be lowest when evaluated at realistic designs. As such, a natural approach is to model $r(\xi)$ using a generative model $p_{\text{ref}}(\xi)$ trained on this data to capture a distribution over designs. In some cases, we might even be able to use a pre-trained foundation model for $p_{\text{ref}}(\xi)$, rather than learning such a model ourselves.

As we expect *higher* misalignment for designs that have *low* probability under the reference distribution over designs $p_{\text{ref}}(\xi)$, we set $r(\xi) = -\log p_{\text{ref}}(\xi)$, yielding the `DiffBED` objective

$$U_{\text{DiffBED}}(\xi) = \alpha \cdot \text{EIG}(\xi) + \log p_{\text{ref}}(\xi). \tag{10}$$

We demonstrate later that such a reference distribution can often be learned from the same data used for learning the likelihood. There are, however, limitations to directly optimising $U_{\text{DiffBED}}(\xi)$, as is standard in BED procedures where $U(\xi) = \text{EIG}(\xi)$. Firstly, it has been shown that the points that deep generative models assign the highest density to are often not themselves reasonable samples (Nalisnick et al., 2018). Secondly, state-of-the-art generative models (e.g. diffusion and flow models) are often implicit, and do not provide direct or reliable access to $p_{\text{ref}}(\xi)$. Thus, rather than directly optimising $U_{\text{DiffBED}}(\xi)$, we instead *sample* from

$$p^*(\xi) \propto \exp(U_{\text{DiffBED}}(\xi)) = p_{\text{ref}}(\xi) \exp(\alpha \cdot \text{EIG}(\xi)), \tag{11}$$

which assigns greater mass near regions of high $U_{\text{DiffBED}}(\xi)$. Sampling $\xi \sim p^*(\xi)$ now ensures that designs are drawn from high-probability regions of $p_{\text{ref}}(\xi)$ while up-weighting those with large EIG. Notably, this approach is compatible with implicit generative design priors, requires no likelihood-dependent training of $p_{\text{ref}}$ nor any modifications to the likelihood, and as such, is compatible with state-of-the-art, pre-trained generative models.

An alternative perspective on Equation (11) is that $p^*(\xi)$ is the unique solution to the following optimisation problem over distributions $q(\xi) \in \mathcal{P}(\Xi)$:

$$p^*(\xi) = \underset{q(\xi) \in \mathcal{P}(\Xi)}{\arg\max} \; \mathbb{E}_{q(\xi)}[\text{EIG}(\xi)] - \frac{1}{\alpha} \text{KL}[q(\xi) \| p_{\text{ref}}(\xi)] \tag{12}$$

where $\alpha$ is the same hyper-parameter that trades off achieving high EIG values with adherence to the reference distribution, as measured by the KL divergence. This solution is often referred to as the *exponentially tilted* distribution (Rawlik et al., 2012).

## 4.2 DiffBED: Diffusion Models as Reference Distributions

Having established $p^*(\xi)$ as our target distribution for producing designs, we now introduce `DiffBED`, our proposed framework which realises this idea using a diffusion model as the reference generative model $p_{\text{ref}}$ in Equation (11). Diffusion models offer state-of-the-art generative quality across images, video, and scientific data. Moreover, their score-based nature naturally enables powerful training-free guidance methods for sampling from tilted distributions (Bansal et al., 2023; Uehara et al., 2025; Ye et al., 2024; Domingo-Enrich et al., 2025; Denker et al., 2024), making them uniquely suited to our framework by allowing sampling from Equation (11) without retraining.

**Information-Guided Diffusion** We aim to sample from the EIG-tilted distribution $p^*(\xi)$, which can be cast as a stochastic optimal control problem (Uehara et al., 2024; Domingo-Enrich et al., 2025). Intuitively, this works by augmenting the SDE in Equation (5) with an additional term $u(\xi_t, t) \, dt$ that seeks to push noisy designs $\xi_t$ towards regions of high EIG during the generative diffusion process.

Specifically, let $s_{\text{ref}}(\xi_t, t)$ denote a score function associated with $p_{\text{ref}}(\xi)$, that is $s_{\text{ref}}(\xi_t, t) = \nabla_{\xi_t} \log p_t^{\text{ref}}(\xi_t)$, where $p_0^{\text{ref}}(\xi_0) = p_{\text{ref}}(\xi_0)$ and each $p_t^{\text{ref}}(\xi_t)$ is subsequently defined by the noising process in Equation (4). This score function could be taken from some pretrained diffusion model on samples of $\xi$, or it can be defined by explicitly defining $p_{\text{ref}}(\xi)$ along with a reference drift vector field $f(x_t, t)$ and noise schedule $g(t)$ that produce a $p_T(\xi_T)$ which is approximately $\mathcal{N}(0, I)$. We can then, in principle, draw samples $\xi \sim p^*(\xi)$ by simulating the SDE

$$d\xi_t = \left[ u(\xi_t, t) + f(\xi_t, t) - g(t)^2 s_{\text{ref}}(\xi_t, t) \right] dt + g(t) \, d\overleftarrow{W}_t \qquad t \in [0, T], \; \xi_T \sim \mathcal{N}(0, I) \tag{13}$$

where the additional drift term $u(\xi_t, t)$ is given by

$$u(\xi_t, t) = g(t)^2 \nabla_{\xi_t} \log \mathbb{E}_{p_{\text{ref}}(\xi_0|\xi_t)} [\exp(\alpha \cdot \text{EIG}(\xi_0))]. \quad (14)$$

This drift $u(\xi_t, t)$ can be interpreted as the gradient of a *value function* that rewards states $\xi_t$ which are likely to yield a high-EIG design $\xi_0$ starting from the current noisy sample $\xi_t$. However, it is intractable due to requiring an expectation over $p_{\text{ref}}(\xi_0|\xi_t)$, and sampling $\xi_0 \sim p_{\text{ref}}(\xi_0|\xi_t)$ would require us to repeatedly solve the SDE (5) starting from the current state $\xi_t$, once per sample.

As such, inspired by recent work on training-free guidance (Chung et al., 2023; Ye et al., 2024), we define $\widehat{\xi}_0(\xi_t) := \mathbb{E}_{p_{\text{ref}}(\xi_0|\xi_t)}[\xi_0|\xi_t]$ and approximate $u(\xi_t, t) \approx g(t)^2 \nabla_{\xi_t}[\alpha \cdot \text{EIG}(\widehat{\xi}_0(\xi_t)]$, which follows from approximating the intractable $p_{\text{ref}}(\xi_0|\xi_t)$ by a delta function located at its mean. While this is a crude approximation to the full distribution, we find that in practice it provides a sufficient guidance signal for obtaining highly informative designs. This approximation is tractable as Tweedie's formula (Robbins, 1956; Efron, 2011; Meng et al., 2021) allows us to exactly calculate $\widehat{\xi}_0(\xi_t)$ in terms of the score function $s_{\text{ref}}(\xi_t, t)$ without needing to simulate the SDE (5). For instance, when $f(\xi_t, t) = -\frac{1}{2}\beta(t)\xi_t$ and $g(t) = \sqrt{\beta(t)}$ (i.e., DDPM (Ho et al., 2020) or the VP-SDE (Song et al., 2021)), Tweedie's formula may be written as $\widehat{\xi}_0(\xi_t) = (\xi_t + (1 - \alpha_t)s_{\text{ref}}(\xi_t, t))/\sqrt{\alpha_t}$, where $\alpha_t = \exp(-\int_0^t \beta(s)\,\mathrm{d}s)$, enabling an efficient approximation of Equation (14) that only requires a single evaluation of the score function.

Putting this together, we obtain an approximate sampler for $p^*(\xi)$ by solving the SDE

$$\mathrm{d}\xi_t = \left[f(\xi_t, t) - g(t)^2 \left(s_{\text{ref}}(\xi_t, t) + \alpha\nabla_{\xi_t}\text{EIG}\left(\widehat{\xi}_0(\xi_t)\right)\right)\right]\mathrm{d}t + g(t)\,\mathrm{d}\overleftarrow{W}_t \qquad t \in [0, T] \quad (15)$$

backwards in time from $\xi_T \sim \mathcal{N}(0, I)$. In practice this simply adds a scaled EIG-gradient estimate to the score network at each step, namely Equation (3) if $y$ is discrete, or some alternative EIG gradient estimator like PCE or NMC if it is not (see e.g. Rainforth et al. (2024, Section 3)).

In sequential settings, DiffBED allows for adaptive design simply by replacing the EIG gradient estimator with an estimator for the incremental EIG, i.e. $\text{EIG}(\xi^k|\mathcal{D}^{k-1})$. Thus it is analogous to the traditional sequential BED approach which chooses designs in a myopic way, except that our optimization for each design uses our guided diffusion approach described above. In Appendix B, we present the full details needed to implement DiffBED (c.f. Algorithm 1).

**Design Sets: Interacting Particle Diffusion**   In applications such as human preference elicitation, each design may itself be a *set* of $S$ elements $\xi = \{\xi^{(1)}, \ldots, \xi^{(S)}\}$ with $\xi^{(j)} \in \mathbb{R}^D$ and $\xi \in \mathbb{R}^{S \times D}$. In such settings, we can leverage a standard diffusion model defined on individual elements, $\xi^{(j)} \in \mathbb{R}^D$, and then set our reference distribution to be an independent mixture of these, i.e., $p_{\text{ref}}(\xi) \propto \prod_{j=1}^S p_{\text{ref}}(\xi^{(j)})$. While sampling a design set, we maintain a noisy set $\xi_t = \left\{\xi_t^{(1)}, \ldots, \xi_t^{(S)}\right\}$, and the reverse process for each element $\xi_t^{(j)}$, for $j = 1, \ldots, S$, is given by

$$d\xi_t^{(j)} = \left[f(\xi_t^{(j)}, t) - g(t)^2\left(s_{\text{ref}}(\xi_t^{(j)}, t) + \alpha\nabla_{\xi_t^{(j)}}\text{EIG}(\widehat{\xi}_0(\xi_t))\right)\right]\mathrm{d}t + g(t)\,\mathrm{d}\overleftarrow{W}_t^{(j)}, \quad (16)$$

where $\widehat{\xi}_0(\xi_t) = \left\{\widehat{\xi}_0(\xi_t^{(1)}), \ldots, \widehat{\xi}_0(\xi_t^{(S)})\right\}$ is the set of element-wise conditional means and $\mathrm{d}\overleftarrow{W}_t^{(j)}$ are independent Brownian increments. Here the diffusion prior contributes an independent score update for each element, ensuring realism, while the EIG term $\nabla_{\xi_t^{(j)}}\text{EIG}(\xi_t)$ introduces cross-element coupling, ensuring informativeness of the entire set as a design. This yields an interacting-particle diffusion that is structurally similar to diversity-oriented particle guidance schemes (Corso et al., 2023; Kirchhof et al., 2025), except that coupling arises from the EIG objective rather than a repulsive kernel designed to enforce diversity.

**Posterior Inference**   For sequential BED, we require a fast, high-fidelity posterior sampler. In our image-scale experiments, a key design choice of DiffBED is to perform inference in a latent space rather than directly in pixel space. This exploits the fact that, in many ostensibly high-dimensional tasks, the information we care about concentrates in a much lower-dimensional representation (e.g., perceptual features), while nuisance variation (background, illumination, pixel noise) can be ignored. This choice also aligns with the recent shift toward *latent world modelling* in areas like robotics (Bar et al., 2025; Garrido et al., 2025; Mei et al., 2026). Operating in this space makes DiffBED robust and scalable, while remaining compatible with problems where the likelihood is naturally defined on top of an encoder. We provide further details of our particle-based inference scheme in Appendix C.

## 5 Related Work

Although the framework of BED has a long history (Lindley, 1956; Bernardo, 1979; Sebastiani & Wynn, 2000), scaling BED to realistic settings remains an open challenge (Rainforth et al., 2024; Huan et al., 2024). Viewing this challenge as one of computational costs, a long list of works (Foster et al., 2019; 2020; Goda et al., 2022; Ao & Li, 2024; Iollo et al., 2025a) have looked to provide improved gradient estimators compared to simple nested Monte Carlo (Rainforth et al., 2018). Notably, Iollo et al. (2025a) also considers the use of diffusion models, but only in an attempt to improve scaling in the *target variable space*, $\theta$, by using a diffusion model for their prior, $p(\theta)$. All of the aforementioned works are restricted to *low–dimensional design spaces*, with the 15-dimensional and 20-dimensional design spaces considered by Iollo et al. (2025b) and Ivanova et al. (2021) respectively being some of the highest dimensional applications of sequential BED. By contrast, we successfully conduct sequential design optimisation in design spaces of over $750\,000$ dimensions.

Model misspecification has also been studied in the context of BED (Feng, 2015; Overstall & McGree, 2022; Go & Isaac, 2022; Forster et al., 2025), with prior works typically looking to address misspecification by explicitly adjusting the model or the EIG itself. We are the first, however, to demonstrate the occurrence of reward hacking–style behaviour and highlight that this is typically unavoidable in high-dimensional design spaces, especially if the likelihood has been learned from data. Sürer et al. (2024); Oliveira et al. (2024) use a Gaussian Process to correct errors in the model, but are focused on the setting of simulator calibration and assume that a dataset of real experimental outcomes is already available prior to doing experimental design. In our high-dimensional, low-data setting, *learning* an explicit bias is not feasible, as it would require large quantities of experimental outcome data, which is fundamentally scarce. Thus, `DiffBED` provides the first scalable solution by instead introducing a design prior to regularise the design optimisation.

Recent BED work has also considered leveraging LLMs to generate natural language questions, e.g. for preference elicitation (Choudhury et al., 2025; Kobalczyk et al., 2025); this represents a problem where the designs are much higher dimensional and more complex than most traditional applications of BED. In general, these approaches work by generating a batch of candidate designs from the LLM, estimating the EIG for each, and then choosing the design with the largest estimate, analogously to the *Rank* variant of `DiffBED` we consider in our experiments. Handa et al. (2024) similarly considers preference elicitation using BED and LLMs, but assumes the parameter and design are supported on an explicit and fixed low-dimensional feature space. Unlike these works, `DiffBED` is explicitly focused on high-dimensional and continuous design spaces, leveraging gradient-based optimization.

## 6 Experiments

We now perform an extensive empirical evaluation of our proposed `DiffBED` method. Although `DiffBED` is not specific to any one experimental setting, our experiments are unified by the theme of human feedback elicitation, as this encapsulates a broad range of important, real-world tasks with high-dimensional designs. In particular, we focus on the setting where designs consist of one or more images. We consider a range of datasets and feedback types, which include rankings and discrete ratings. We defer in-depth experimental details, including additional results and ablations, to the Appendix.

**Baselines** Our primary baseline is the current standard paradigm for *BED*, namely, direct gradient-based maximisation of an EIG estimator (Huan et al., 2024; Rainforth et al., 2024). We also compare against *Entropy*, a variant of `DiffBED` where we guide the diffusion model with the marginal predictive entropy rather than the EIG. We further consider two ranking baselines. *Rank (Pool)* estimates the EIG for 1000 candidate designs per iteration (chosen to roughly match the runtime of `DiffBED`) and selects the highest EIG candidate. To make this baseline as strong as possible, candidates are drawn directly from the data used to train the diffusion model, rather than being generated. *Rank (Diffuse)* instead samples candidates from $p_{\text{ref}}(\xi)$ via the diffusion model, as required when no pool is available. Under compute normalisation this corresponds to $N \approx 4$ candidates (Appendix B.3); we use $N = 5$ to conservatively allocate more compute to *Rank (Diffuse)*. Finally, we compare against *Random*, a simple baseline where designs are selected uniformly at random from a set of feasible designs.

**Metrics** To evaluate the different design strategies, we track our ability to recover a ground truth target variable $\theta_{\text{true}}$. For the location-finding task, we report the mean $L_2$ error between the ground truth $\theta_{\text{true}}$ and samples from the current posterior over $\theta$. For our image-space experiments, we instead report the mean cosine similarity between $\theta_{\text{true}}$ and samples from the current posterior. We

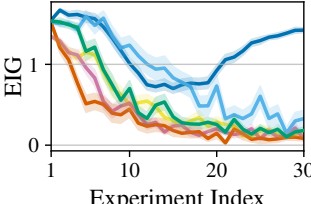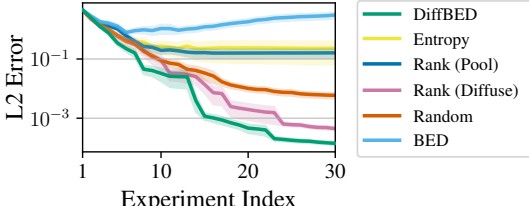

Figure 3: Incremental EIG (under $p_{\mathrm{model}}$) at each experiment iteration (left) and average $L_2$ error between posterior samples and ground truth locations (right) for the source location finding experiment. `DiffBED` achieves the lowest $L_2$ error, with traditional BED and the rank methods all doing poorly.

also evaluate the incremental EIG of the chosen design at each step, but emphasise that this should not be viewed as a success metric itself but rather an insightful quantity to track. We additionally include several qualitative results (c.f. Figures 1 and 2). Results on source location finding and MNIST are averaged over 25 random seeds, while CelebA and Zappos are averaged over 10.

## 6.1 SOURCE LOCATION FINDING

Our first experiment is source location finding, a synthetic problem widely studied in BED (Ivanova et al., 2021; Blau et al., 2022; Iollo et al., 2025a). The goal is to infer the location of $N$ sources $\theta = (\theta^{(1)}, \theta^{(2)}, ..., \theta^{(N)}) \in \mathbb{R}^{N \times D}$, each of which emits a signal that decays according to an inverse square law. At each experiment, a sensor placed at a location $\xi \in \mathbb{R}^D$ records a noisy measurement $y \in \mathbb{R}$ of the signal intensity at $\xi$. The (noiseless) signal strength is given by $\mu(\theta, \xi) = b + \sum_{i=1}^{N} \frac{\alpha_i}{m+\|\theta^{(i)}-\xi\|^2}$, where $\alpha_i$, $b$, and $m$ are known constants. The observed signal follows a log-normal distribution $\log y \mid \theta, \xi \sim \mathcal{N}(\log \mu(\theta, \xi), \sigma)$. We use $p_{\mathrm{model}}(y \mid \theta, \xi)$ to represent this assumed likelihood.

As this problem does not use a learned likelihood, we explicitly introduce model misalignment to illustrate the utility of `DiffBED`. We assume that there is a region of design space $\Xi^* \subseteq \Xi$ which is *well-aligned*, i.e., where $p_{\mathrm{model}}$ captures the data generating process. Outside of this region, we assume that data is drawn according to an out-of-distribution likelihood $p_{\mathrm{ood}}(y \mid \theta, \xi)$, leading to the mixture $p_{\mathrm{true}}(y|\theta, \xi) = \mathbf{1}_{\Xi^*}(\xi)p_{\mathrm{model}}(y|\theta, \xi) + (1 - \mathbf{1}_{\Xi^*}(\xi))p_{\mathrm{ood}}(y|\theta, \xi)$. We take $\Xi^*$ to be a collection of eight disjoint circles. For $p_{\mathrm{ood}}$, we use the same form as $p_{\mathrm{model}}$, except with different choices for $\alpha$, leading to a systematically lower signal mean when $\xi \notin \Xi^*$. Inference is done using $p_{\mathrm{model}}$. The reference $p_{\mathrm{ref}}(\xi)$ is taken to be a mixture of Gaussians, centred on $\Xi^*$. See Appendix D.

As seen in Figure 3, BED produces designs with high incremental EIG estimates, but it often produces designs outside of $\Xi^*$. This leads to out-of-distribution data, hampering inference and resulting in an $L_2$ error which is *worse* than a random draw from $p_{\mathrm{ref}}(\xi)$. Interestingly, *Rank (Pool)* considers far more candidates than *Rank (Diffuse)* (1000 vs. 5), yet achieves worse $L_2$ error. This is because *Rank* greedily selects the highest EIG candidate under $p_{\mathrm{model}}$, and a larger set of candidates is more likely to contain designs that are unlikely under $p_{\mathrm{ref}}$, again yielding out-of-distribution data. This mirrors a phenomenon in active learning, where the performance of BALD can degrade as the candidate pool grows, partly because larger pools contain more unlikely inputs (Bickford Smith et al., 2023). Further, as designs are greedily chosen based on a noisy estimate of the true EIG, larger pools exacerbate "winner's curse" effects: the top-ranked design is increasingly likely to be chosen due to estimator noise rather than genuine informativeness. The cumulative effect of these pathologies is seen also in the EIG, where the incremental EIG *increases* interestingly starts to increase again as the posterior becomes degenerate at later experiment iterations.

These pathologies are mitigated by `DiffBED`. Although `DiffBED` attains lower incremental EIG under $p_{\mathrm{model}}$ during optimisation, it directly generates designs in $\Xi^*$ rather than selecting from an increasingly large candidate pool, effectively guarding against misalignment. This translates into substantially better learning, with `DiffBED` recovering the true source locations to high precision.

## 6.2 INFORMATION-THEORETIC SEARCH

We now evaluate `DiffBED` on high-dimensional, information-theoretic search tasks, where the goal is to recover a ground-truth image based on user feedback. Here, designs are *sets* of images and the outcomes $y$ are rankings indicating the relative similarity of each design to a ground-truth image.

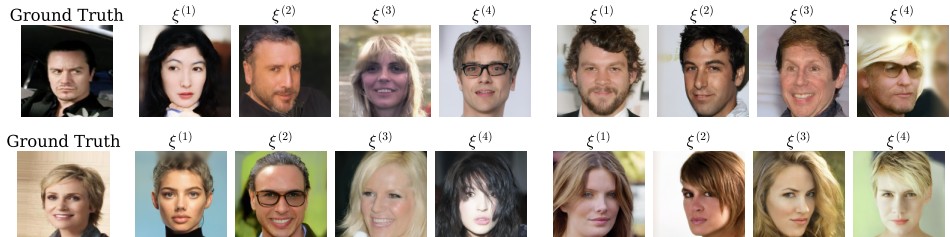

Figure 4: Design sets of four images generated by `DiffBED` for CelebA Search, against two ground truths (top and bottom row), at experiment iteration 1 (centre, both rows) and 40 (right, both rows).

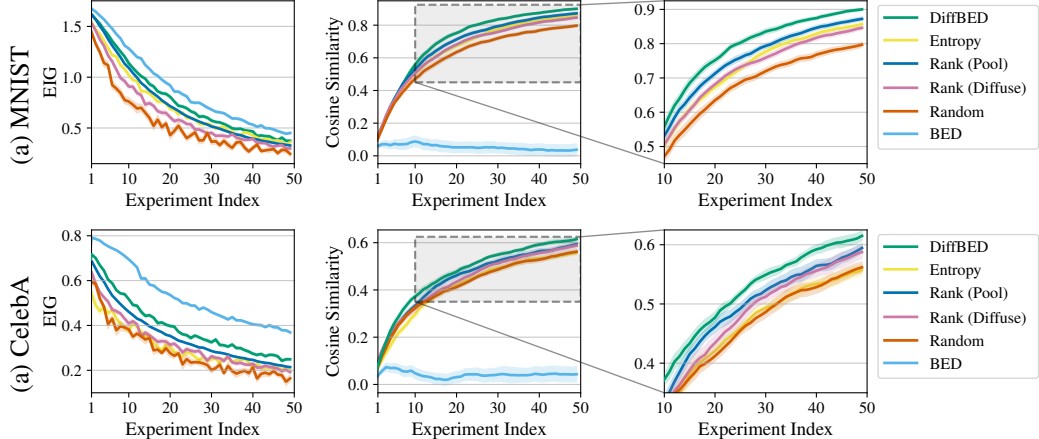

Figure 5: EIG and cosine similarities for search (mean with $\pm$ std. error shading), where designs are sets of four images and responses are the rank of the top-two candidates. `DiffBED` achieves the highest mean cosine similarity, while standard BED fails to solve the task despite achieving high EIGs.

As a motivating example, suppose an eyewitness of a crime is being interviewed in order to construct possible images of the suspect that we wish to be perceptually close to the true suspect. Though it is not generally possible for the eyewitness to directly generate accurate images, they likely can positively identify a photo of the true suspect if shown one and more generally provide feedback when shown images. We can therefore instead iteratively show the eyewitness sets of candidate images $\xi = (\xi^{(1)}, \ldots, \xi^{(S)})$ and have them provide feedback by ranking the images by how well they match the suspect. Such an approach has historically been deployed by UK police forces, but with software that uses low-resolution images and chooses them in a heuristic manner (VisionMetric, 2019).

To apply BED, we require a model approximating the complex relationship between the design $\xi$ and the ranking $y$. Since a person's perception of images does not operate at a pixel-by-pixel resolution, we can assume a model $p_{\mathrm{model}}(y \mid \theta, \xi)$ where $\theta$ is a rich, sufficiently high-dimensional feature space encoding of an image and our likelihood is based around the similarity of each $\xi^{(j)}$ to an underlying $\theta_{\mathrm{true}}$. For our experiment setup, a simulated participant is given a set of $S = 4$ images and their response, $y$, is a ranking of the top $M = 2$ images. We evaluate `DiffBED` on MNIST and CelebA (LeCun et al., 1998; Liu et al., 2015), where $\theta$ is obtained using SimCLR embeddings (Susmelj et al., 2020; Chen et al., 2020) for the former and a pre-trained VGGFace2 model (Cao et al., 2018) fine-tuned using a triplet loss for the latter. Although we do inference in an embedding space, these embeddings can be decoded to yield posterior samples in image space. See Appendix A and Appendix C.

We plot the EIG and cosine similarities between posterior samples and $\theta_{\mathrm{true}}$ in Figure 5. While standard BED obtains high EIG under $p_{\mathrm{model}}$, model misalignment leads to designs that are indistinguishable from pure noise (Figure 1). Thus, the given preferences are meaningless, yielding a near-zero cosine similarity throughout. In contrast, `DiffBED` produces *realistic* designs (Figure 4) with high EIG. Although `DiffBED` obtains a lower EIG than standard BED, our method deliberately trades off EIG for realism. This enables `DiffBED` to effectively determine $\theta_{\mathrm{true}}$, achieving the highest cosine similarity. Notably, `DiffBED` outperforms *Rank (Pool)* despite only generating a single

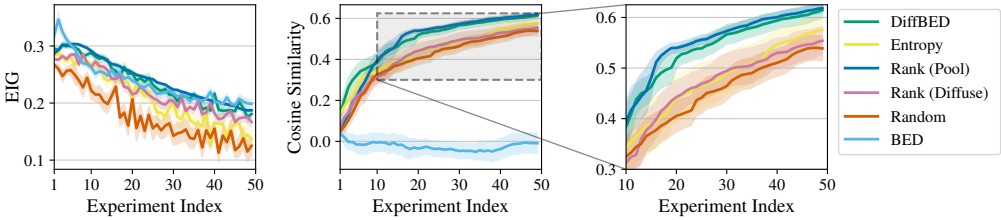

Figure 6: Example high resolution designs produced by `DiffBED` on the Zappos dataset.

Figure 7: Search on the large-scale Zappos dataset with high-resolution ($512 \times 512$) images. Designs are single images with discrete ratings as responses. Even at this scale, `DiffBED` remains effective.

design set at each iteration, whereas *Rank (Pool)* considers $1\,000$ candidate design sets. Qualitatively, `DiffBED` first explores diverse, coarse attributes (e.g., gender, hair colour), before refining to more nuanced details in later iterations (Figure 4). This is in contrast to predictive entropy (*Entropy*) which does not in any way encourage the set of images the user is asked to rank to be different, with the EIG needed to capture the nuance that the rankings need to be informative, not simply uncertain.

### 6.3 TEXT-TO-IMAGE FOUNDATION MODELS

We now scale `DiffBED` even further by leveraging text-to-image foundation models as $p_{\mathrm{ref}}$, focusing on the problem of preference elicitation for e-commerce products. Specifically, we consider a setup where the user is shown a single image of a shoe and asked to give a rating of $1$ to $5$, using this to hone in on the users notion of an "ideal shoe" that we can use for making recommendations. For $p_{\mathrm{ref}}$ and $s_{\mathrm{ref}}$, we use Stable Diffusion v1.5 (Rombach et al., 2022), a 1B parameter foundation model, fine-tuned on the Zappos (Yu & Grauman, 2014) dataset of high resolution ($512 \times 512$) images of shoes. Ratings are modelled by the Ordinal Logit Model which assumes the discrete score is correlated with an underlying reward proportional to the similarity of the design image to the image of the user's ideal shoe. See Appendix A.3.3.

Figure 7 shows that standard BED again fails to learn anything meaningful, while `DiffBED` produces highly informative designs that are also highly realistic (Figure 6). Overall, `DiffBED` outperforms all baselines except *Rank (Pool)*, and even there the gap is only marginal. The performance gap between `DiffBED` and *Rank* variants will generally widen when the set of sub-designs that need to be chosen (e.g. sets of images to be compared) grows, noting the relatively weaker performance of *Rank* seen in 6.2 where designs are *sets*. In such cases, `DiffBED` leverages gradient information to directly generate informative design sets, whereas Rank must evaluate many random combinations to (by chance) identify an informative one. As such, we expect `DiffBED` to transfer naturally to batch design, where multiple experiments are designed at each stage by optimising a joint EIG objective (Rainforth et al., 2024).

### 7 CONCLUSION

We have shown that model misalignment is a key barrier in scaling BED to high-dimensional design spaces, and that when the likelihood is unreliable, optimising standard information-theoretic objectives can induce reward-hacking behaviour, producing meaningless designs. To address this, we introduced `DiffBED`, which explicitly trades off EIG against design realism, captured by a reference design distribution. We actualise this by guiding a diffusion model with an information-theoretic objective, steering sampling towards designs that are both realistic and highly informative. By mitigating the exploitation of model misalignment, `DiffBED` enables BED with image-scale designs for the first time. Our results on challenging preference elicitation tasks demonstrate that explicitly incorporating design realism opens the door to applying BED in complex domains previously beyond its reach.

ACKNOWLEDGMENTS

AS is supported by the EPSRC CDT in Statistics and Machine Learning (EP/Y034813/1). RK is supported by the Martingale Postgraduate Foundation. GK and TR are supported by the UK EPSRC grant EP/Y037200/1.

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

APPENDIX CONTENTS

# A  LIKELIHOOD MODELS

This section provides details for all likelihood models studied in our experiments (Section 6).

When picking an experiment paradigm, practitioners must consider the trade-off between the information conveyed in a given observation (e.g. binary preference, partial ranking), and the cost of running said experiment. For example, a full-ranking contains high amounts of information per observation, but may not be feasible to collect. We summarize the paradigms studied in our experiments in Table 1.

Table 1: Common paradigms for eliciting human feedback.

| Paradigm | Example design | Example feedback | Example Likelihood |
|---|---|---|---|
| **Binary Ranking** (Appendix E) |  | "Image 1 ≻ Image 2" | Bradley–Terry (BT) (Bradley & Allan, 1952) |
| **Ranking $k$ designs from $n$** (Section 6.2) |  | "Image 1 ≻ Image 2 ≻ Image 3" | Plackett–Luce (PL) (Plackett, 1975; Luce, 1959) |
| **Discrete rating** (Section 6.3) |  | "4 out of 5 stars" | Ordinal Logit (OL) (Mccullagh, 1980) |

## A.1  LIKELIHOOD PMFS

We now provide the functional form of the likelihood models considered in our experiments. Note that all of the models leverage a latent reward model, $r_\theta(\xi)$, parametrized by the quantity of interest, $\theta$, which assigns a score $r_\theta(\xi) \in \mathbb{R}$ to the design/each element in the design set.

**Binary Preferences: Bradley-Terry (Bradley & Allan, 1952)**

Let $\xi = \{\xi^{(1)}, \xi^{(2)}\}$ be a design set of two items. The Binary Bradley-Terry model assumes,

$$p\Big(y = \xi^{(i)} \succ \xi^{(j)} \mid \theta, \xi\Big) = \frac{\exp\big(r_\theta(\xi^{(i)})\big)}{\exp\big(r_\theta(\xi^{(i)})\big) + \exp\big(r_\theta(\xi^{(j)})\big)}.$$

**Partial Rankings: Plackett-Luce (Plackett, 1975)**

Let $\xi = \{\xi^{(1)}, \xi^{(2)}, \ldots, \xi^{(S)}\}$ be a design set of $S$ items. Suppose we observe a partial ranking of the form

$$\xi^{(\sigma_1)} \succ \xi^{(\sigma_2)} \succ \cdots \succ \xi^{(\sigma_M)}, \quad \text{with } M \leq S,$$

where $\sigma = (\sigma_1, \sigma_2, \ldots, \sigma_M)$ is an ordered list of distinct indices indicating the ranked items. Then the Plackett–Luce likelihood of the observed partial ranking is

$$p\Big(y = \xi^{(\sigma_1)} \succ \xi^{(\sigma_2)} \succ \cdots \succ \xi^{(\sigma_M)} \mid \theta, \xi\Big) = \prod_{j=1}^{M} \frac{\exp\big(r_\theta(\xi^{(\sigma_j)})\big)}{\sum\limits_{\xi^{(i)} \in C_j} \exp\big(r_\theta(\xi^{(i)})\big)},$$

where $C_j$ is the set of items available at stage $j$, with $C_1 = \xi$ and $C_{j+1} = C_j \setminus \{\xi^{(\sigma_j)}\}$.

**Discrete Ratings: Ordinal Logit (Mccullagh, 1980)**  Let $\xi$ being a single item, unlike the Bradley-Terry and Plackett-Luce models, which operate on design sets. Under the Ordinal Logit model, observations are one of $K$ ordered, discrete categories, modelled under the following PMF:

$$p(y = k \mid \xi, \theta) = \sigma\left(\tfrac{b_k - r_\theta(\xi)}{\tau}\right) - \sigma\left(\tfrac{b_{k-1} - r_\theta(\xi)}{\tau}\right), \qquad k = 1, \ldots, K,$$

with $\sigma(x) = (1 + e^{-x})^{-1}$.

### A.2 Parametrisation of Latent Reward Model

As reiterated in the main body of the paper, in the high-dimensional, challenging settings considered in the paper, defining a likelihood, $p(y|\theta, \xi)$, that mimics human behaviour for all $\{\theta, \xi\}$ is extremely challenging. Further, we note that $p(y|\theta, \xi)$ can often not be directly trained in a supervised manner, and practitioners must leverage domain-specific insight. This is especially true in cases in which the $\theta$ of interest is inherently latent, e.g. user preferences. As such, we must often assume the structure of the canonical models. However, in all of the models considered above, we must still define a latent reward model, $r_\theta(\xi)$.

In many problems, e.g. preference-elicitation, the interpretable features that are most pertinent can be identified. In such settings, the following parametrization can be considered: $r_\theta(\xi) = \theta^T d(\xi) = \sum_{k=1}^{K} \theta_k d_k(\xi)$, of $K$ interpretable features, e.g. alignment to a style. In this parametrisation, $\theta$ can be viewed as interpretable preference weights for the $K$ different aspects. In other problems, where such interpretable features can not be identified, one can opt for the following parametrisation: $r_\theta(\xi) = \cos\_\text{sim}(\theta, d(\xi))$ where $d(\cdot)$ is an encoder of $\xi$ trained in an unsupervised fashion.

A further practical benefit of these parametrisations is that they are data-efficient, arising from the incorporation of task-specific knowledge through the fixed, pre-trained feature extractors, $d(\cdot)$, which should support faster learning. However, we stress that `DiffBED` is agnostic to the parametrisation of the latent reward model, $r_\theta(\cdot)$, and more generally, the response paradigm and likelihood model used.

**Temperature hyper-parameter, $\tau$**  In line with common practice, we restrict the range of the latent reward, $r(\xi)$, to a pre-defined range $[-1, 1]$, for example through normalisation or by computing the reward as cosine similarities, before introducing a multiplicative scaling parameter, $\tau$, to expand the range to $\left[-\tau^{-1}, \tau^{-1}\right]$. This enables explicit control over the sharpness of the assumed likelihood model's response distribution. As $\tau^{-1} \to 0$, the distribution over potential observations for any design across all the models introduced tends to a uniform distribution. On the other hand, increasing $\tau^{-1}$ results in peakier likelihoods, and faster shrinkage of the posterior. However, as it assigns a stronger belief on the data generating mechanism, if this does not reflect reality, the degree of misalignment with the true data generating process is also increased.

### A.3 DETAILS OF LATENT REWARD MODELS

In this section, we present the specific details of the latent reward models used in our image experiments. In the search experiments, we leverage unsupervised encoders trained through contrastive losses, to encourage learning general representations of the data. In the preference elicitation experiment, we leverage an interpretable feature extractor.

#### A.3.1 MNIST SEARCH (SECTION 6.2)

For the MNIST search experiment, we train an encoder using the SimCLR loss (Chen et al., 2020). We leverage a simple CNN architecture, with two convolutional layers and fully-connected layer. The embedding dimensionality is $K = 32$, with a projection-head of size $P = 512$, a hyper-parameter for computing SimCLR contrastive losses. The encoder is trained for $50\,000$ steps, with a batch size of $1\,024$. The underlying reward function used in our Bradley-Terry model is $r_\theta(\xi) = \tau^{-1}\mathrm{cosine\_sim}(\mathrm{simclr}(\xi), \theta)$, where $\mathrm{simclr}$ is our encoder. We take $\tau^{-1}$ to be 10 in Figure 5.

To explicitly incorporate model misspecification into the responses, we leverage a pre-trained discriminator that detects out-of-distribution (OOD) MNIST images. For in-distribution images, the simulated observation $y$ is from a re-normalised PMF of the rankings that don't include any OOD images, and in cases where all images are classed as OOD, the ranking observations are generated uniformly at random.

#### A.3.2 CELEBA

**Search (Section 6.2)** We train a CelebA encoder on the CelebA train set. We take a pre-trained VGGFace2 model as our backbone (Cao et al., 2018), and fine-tune it. We remove the original last layer and replace it with a 64 dimensional linear last layer, hence $\theta$ is, 64 dimensional. We freeze the backbone, training only the last layer weights, before unfreezing the final block before the last layer, and fine tuning these weights alongside the last layer's weights. We train using a triplet loss, since CelebA includes identity labels. The underlying reward function used in our Placket-Luce model is $r_\theta(\xi) = \tau^{-1}\mathrm{cosine\_sim}(\mathrm{vgg}(\xi), \theta)$, where $\mathrm{vgg}$ is our fine-tuned encoder. We take $\tau^{-1}$ to be 10. We run the experiment for 50 iterations.

**Preference Elicitation (Appendix E)** We train a supervised feature extractor on the CelebA dataset by fitting a multi-label attribute classifier to predict the following $C = 23$ characteristics that have binary labels in the dataset:

```
Bald, Wavy_Hair, Straight_Hair, Receding_Hairline,
Bangs, Sideburns, Black_Hair, Gray_Hair, Blond_Hair,
Brown_Hair, No_Beard, 5_o_Clock_Shadow, Mustache, Goatee,
Big_Lips, Big_Nose, Eyeglasses, Smiling, Heavy_Makeup,
Wearing_Lipstick, Wearing_Necklace, Wearing_Earrings.
```

We leverage a ResNet50 model initialised with ImageNet weights, with the final layer removed and replaced with a linear layer that maps to $C = 23$. The model is trained for 25 epochs, with a batch size of 128. We use binary cross-entropy with logits, assigning per-attribute positive weights equal to the negative-to-positive sample ratio to correct for class imbalance, and Adam as the optimizer.

We use a Bradley-Terry model, with underlying reward being $r_\theta(\xi) = \tau^{-1} \sum_{i=1}^{N} d_i(\xi) \cdot \theta_i$, where $d(\xi) \in \mathbb{R}^{|C|}$ is a vector with element $d_i(\xi)$ being the ResNet50 classifier probability that attribute $i$ is present in a design image.

#### A.3.3 ZAPPOS SEARCH (SECTION 6.3)

We train a Zappos encoder using the SimCLR loss, with $K = 64$ and $P = 512$. We leverage a ResNet50-based model and initialized with torchvision's retrained ImageNet-1K (Deng et al., 2009) weights[1], with the final fully connected (FC) layer removed and replaced by a linear projection to a $K$ embedding space. The model is trained 50,000 steps, with a batch size of 256, a learning-rate

---

[1]https://pytorch.org/vision/stable/models.html

---

**Algorithm 1** `DiffBED`: BED with Information Guided Diffusion

---

**Input: BED setup:** prior $p(\theta)$; likelihood $p(y \mid \theta, \xi)$; experiment steps $K$
**Input: Diffusion:** reference score model $s_\varphi^{\text{ref}}$; SDE steps $T$; guidance scale $\alpha$
**Input: Inference:** particle count $N$; particle filter;

**Output:** Designs $\xi^{1:K}$, observations $y^{1:K}$, particles $\{\theta_n^{(K)}\}_{n=1}^N$;

1: **Initialize:** draw $\{\theta_n^{(0)}\}_{n=1}^N \sim p(\theta)$; set $\xi^{1:0} \leftarrow \emptyset$, $y^{1:0} \leftarrow \emptyset$.
2: **for** $k = 1, \ldots, K$ **do** ▷ Sequentially design and run each experiment
    — **Diffusion-based design sampling** —
3:    $\xi_T \sim \mathcal{N}(0; I)$ ▷ Initialize at noise
4:    **for** $t = T, T-1, \ldots, 1$ **do**
5:        $g_t \leftarrow \nabla_{\xi_t} \widehat{\text{EIG}}\big(\hat{\xi}_0(\xi_t)\big)$ ▷ Estimate EIG gradient using $\{\theta_n^{(k-1)}\}$
6:        $\xi_{t-1} \leftarrow \text{SDEstep}(\xi_t, s_\varphi^{ref}(\xi_t, t), g_t, \alpha)$ ▷ SDESolver step
7:    **end for**
8:    Set design $\xi^k \leftarrow \xi_0$
    — **Run experiment and update posterior** —
9:    $y^k \leftarrow y \sim p(y \mid \theta^\star, \xi^k)$
10:    $\xi^{1:k} \leftarrow \xi^{1:k-1} \cup \{\xi^k\}$, $y^{1:k} \leftarrow y^{1:k-1} \cup \{y^k\}$
11:    $\{\theta_n^{(k)}\} \leftarrow \text{ParticleFilter}\big(\{\theta_n^{(k-1)}\}, \xi^{1:k}, y^{1:k}\big)$
12: **end for**
13: **return** $\xi^{1:K}, y^{1:K}, \{\theta_n^{(K)}\}$

---

of 1, and with SGD as the optimizer. We utilise an Ordinal-Logit model with an underlying reward $r_\theta(\xi) = \tau^{-1} \cos\_\text{sim}(\text{simclr}(\xi), \theta)$.

# B    DiffBED Details

This section contains the algorithmic details needed to implement `DiffBED` in practice. In Algorithm 1, we provide pseudocode which describes how `DiffBED` is applied end-to-end for (sequential) BED problems.

## B.1   Reference Models

For each experiment, we require a reference diffusion model $p_{\text{ref}}(\xi)$ whose samples produce reasonable designs for the problem at hand. We detail our specific choices for each dataset here.

**MNIST**   We use the training script provided in the PyTorch codebase provided by Song et al. (2021)[2] to train an unconditional MNIST diffusion model. We use an NCSN++ UNet (64 base channels, one residual block per level) with VP-SDE with a linear noise schedule $\beta(t) = \beta_{\min} + t(\beta_{\max} - \beta_{\min})$ $t \in [0, 1]$, i.e., $f(x, t) = -\frac{1}{2}\beta(t)x$ and $g(t) = \sqrt{\beta(t)}$ ($\beta_{\min} = 0.01, \beta_{\max} = 20$). We train for 500k iterations, with EMA (0.999) and a batch size of 256, using the Adam optimizer ($lr = 0.0002$, with gradient clipping at norm= 1.0).

**CelebA and Zappos**   For the higher dimensional datasets, CelebA (Liu et al., 2015) and Zappos (Yu & Grauman, 2014), we leverage the Hugging Face `diffusers` library[3], which provides standardized pipelines for training, inference, and sampling of diffusion and latent diffusion models. For CelebA, we use a pre-trained latent diffusion model (Rombach et al., 2022) checkpoint.[4] For Zappos, we use a fine-tuned version of Stable Diffusion v1.5 (SDv1.5). [5]

---

[2] https://github.com/yang-song/score_sde_pytorch
[3] https://huggingface.co/docs/diffusers
[4] https://huggingface.co/CompVis/ldm-celebahq-256
[5] https://huggingface.co/benisonjac/finetune-of-stable-diffuson-on-Zappos-shoe-dataset

## B.2 SAMPLING

We now turn to the details involved in sampling from $p^*(\xi)$ using Equation (15). In particular, Shen et al. (2024) considers the shortfalls of training-free guidance of diffusion models and leverage ideas from optimization literature to mitigate them. We find that Polyak step-size parametrisation of the guidance scale is beneficial in finding a favourable trade-off between the informativeness and realism of designs. Namely, this considers a time-dependent guidance scale as a multiplier of the EIG gradient estimator, which we denote as $\gamma_t$ for brevity, during the reverse-process:

$$\alpha(t) = \eta \cdot \frac{\|\epsilon_\varphi(\xi_t, t)\|}{\|\gamma_t\|_2^2}. \tag{17}$$

Across all experiments, we use the Euler–Maruyama discretization of the reverse SDE, equivalent to the ancestral/DDPM sampler. We provide further dataset-specific sampling hyper-parameters below. We use uniform time-steps on all datasets, with 500/250/100 steps on MNIST/CelebA/Zappos, respectively. The choice of the $\eta$ is an empirical one, determined by the robustness of the reference diffusion model (Ye et al., 2024). We set this parameter by visually inspecting a small set of samples for increasing values of $\eta$, setting the maximal value that still consistently produces high-fidelity images. We use $\eta = 0.0375, 0.10$ for CelebA/Zappos experiments, and present example samples produced at this guidance scale in Figure 11a and Figure 6.

As SDv1.5 is a text-conditioned model, we use the following prompt to capture the data-distribution of interest for the Zappos task: ``Studio product photo of a footwear, isolated on white background, high detail''. To avoid artefacts, we also use the following negative prompt: ``blurry, low resolution, watermark, deformed''.

## B.3 NUMBER OF FUNCTION EVALUATIONS (NFEs)

We report cost in terms of *network function evaluations* (NFEs), counting a single forward pass through either the diffusion model or the design-embedding network as one NFE. While the embedding network is substantially cheaper than the diffusion model in our experiments, we ignore this difference, which favours the *Rank (Diffuse)* baseline, and treat both as unit-cost for simplicity.

**DiffBED** At each reverse-diffusion step we (i) evaluate the diffusion score (1 NFE), and (ii) estimate the EIG, which in our image experiments is dominated by a single forward pass through the design-embedding network (1 NFE). To form the information-guidance term, we additionally back-propagate through both networks; we approximate the cost of each backward pass as $2\times$ its corresponding forward pass. Therefore, over the reverse process, we incur $4K$ NFEs.

*Rank (Diffuse)* **Baseline** With $N$ candidate designs, this baseline requires running a $K$-step reverse process for each candidate ($KN$ NFEs), plus an additional EIG estimate per candidate ($N$ NFEs), for a total of $KN + N$ NFEs.

**Compute Normalization** When comparing *Rank (Diffuse)* to DiffBED, we match compute by choosing $N$ such that $KN + N \approx 4K$ (for large $K$, this is well-approximated by $N \approx 4$). In practice, we use $N = 5$ (rather than $N \approx 4$) to err on the side of *over*-allocating compute to the baseline.

## C POSTERIOR INFERENCE

We now present details on how we conduct inference on the embedding space during active experimentation. In search problems, as the encoders are trained with a cosine-similarity objective, both our likelihoods and the geometry of the problem depend only on the *direction* of $\theta$, not its scale. Similarly, in preference elicitation settings, we often want to learn normalized preference weights [6], in order to allow for explicit control and regularisation over the sharpness of the preference distributions, for example through a scaling hyper-parameter.

In such settings, although $\theta$ lives in $\mathbb{R}^D$, the unit-length constraint removes one degree of freedom, leaving an intrinsic dimension of $D - 1$. Accordingly we place a *uniform prior* on the unit sphere $\mathbb{S}^{D-1} = \{\theta \in \mathbb{R}^D : \|\theta\|_2 = 1\}$ the $(D-1)$-dimensional manifold of $D$-dimensional vectors of length one. We then explicitly approximate the posterior on the unit sphere.

---

[6]Another alternative is to learn weights that sum to one.

**Particle-Based Inference**  At the start of the experimentation we draw an i.i.d. set of particles $\{\theta_i^{(0)}\}_{i=1}^N \sim \mathrm{Unif}(\mathbb{S}^{D-1})$ to represent this prior. At each experiment index $k$, after observing an outcome $y_k$ at design $\xi_k$, we update particle weights according to the likelihood

$$\log w_i^{(k)} = \log w_i^{(k-1)} + \log p(y^k \mid \theta_i^{(k-1)}, \xi^k).$$

Weights are normalized and particles are resampled via multinomial resampling to prevent degeneracy (Doucet et al., 2001), yielding an empirical posterior approximation $\{\theta_i^{(t)}\}$.

To maintain diversity and ensure that the particle cloud accurately tracks the true posterior on the sphere, we rejuvenate the particles by applying a *projected unadjusted Langevin algorithm* (ULA) on $\mathbb{S}^{D-1}$:

$$\theta \leftarrow \mathrm{Normalize}\Big(\theta + \frac{\epsilon}{2} P_\theta \nabla_\theta \log \pi_k(\theta) + \sqrt{\epsilon}\, P_\theta \eta\Big),$$

where $P_\theta = I - \theta\theta^\top$ projects gradients and noise onto the tangent space $T_\theta \mathbb{S}^{D-1}$ and $\eta \sim \mathcal{N}(0, I)$. This step can be viewed as an unadjusted discretization of the Riemannian Langevin diffusion on $\mathbb{S}^{D-1}$ (Girolami & Calderhead, 2011; Patterson & Teh, 2013), where the drift term is the gradient of the *full current posterior* $\pi_k(\theta) \propto \pi_0(\theta) \prod_{s=1}^k p(y^s \mid \theta, \xi^s)$, so that each move incorporates all experimental observations up and including experiment index $k$.

**Computational Cost**  As the log-likelihoods are parametric functions of simple vector products, i.e. cosine-similarities, between $\theta$ and encodings of previous designs, both resampling and Langevin steps are very efficient.[7] Crucially, resampling and Langevin steps do not require expensive neural network evaluations per $\theta$ particle. For all experiments, we leverage $N = 10^6$ particles, take 100 Langevin steps, and have step-size $\epsilon = 10^{-4}$. Note that the cost of storing $N = 10^6$ particles of dimensionality $D = 32, 64$ has costs $\approx 122, 244$ MiB of memory, which is a negligible overhead on modern GPUs.

**Decoding from Latent Space**  We can also recover $\theta$ in full image space from latent posterior samples. The simplest approach is nearest-neighbour retrieval from a large pool. Instead, to obtain the posterior samples shown in Figure 2, we guide a diffusion model over $\theta$ (which is also image-valued in our search experiments) to target the distribution

$$p^*(\theta) \propto p(\theta) \exp\big(\alpha \cdot \mathrm{cos\_sim}(\theta_{\mathrm{latent}}, E(\theta))\big). \tag{18}$$

Intuitively, this synthesizes image-space $\theta$ whose embedding $E(\theta)$ matches a latent posterior particle $\theta_{\mathrm{latent}}$ under cosine similarity. To sample from $p^*(\theta)$ in Equation 18, we use an approach that is directly analogous to that in Section 4.2, now guiding the diffusion model to sample $\theta$.

# D  LOCATION FINDING

In this section we present additional details for our source location finding experiment in Section 6.

**Interpretation of Data-Generating Mechanism**  In many real-world source-finding tasks, such as localizing radio beacons, chemical or radiation leaks, or acoustic emitters, the assumed physical model (e.g., an inverse-square decay) is only accurate in regions where the environment is simple and well-behaved. In such areas, such as open or unobstructed areas, the measurements should typically follow the expected model, $p_{\mathrm{model}}$. However, when the sensor moves into more complex environments, such as areas with obstacles, turbulent airflow, or reflecting surfaces, the underlying physics changes, producing systematically different observations. Our mixture formulation captures this spatially dependent model fidelity by allowing designs inside certain regions to follow the assumed model, while those outside follow an alternative, unknown likelihood, $p_{\mathrm{ood}}$. Effectively modelling the physics outside of the simple case where $p_{\mathrm{model}}$ is accurate may not be feasible. If this is the case, we may wish to find informative design, whilst encouraging adherence to a design prior, where we can trust our ability to model the data-generating process.

**Hyperparameter Settings**  Following Foster et al. (2021), we set $D = 2$, $C = 2$, $\alpha_1 = \alpha_2 = 1$, $m = 10^{-4}$, $b = 10^{-1}$, $\sigma = 0.25$, and design $K = 30$ sequential design optimizations. For `DiffBED`

---

[7]Note that embeddings of designs/design sets can be cached, and as such, just needs to be computed once. For the rest of the roll-out, it can be shared across all particles for resampling and Langevin-step computation.

and Entropy, as with MNIST, we use a VP-SDE with a linear noise schedule $\beta(t) = \beta_{\min} + t(\beta_{\max} - \beta_{\min})$ $t \in [0, 1]$, i.e., $f(x, t) = -\frac{1}{2}\beta(t)\,x$ and $g(t) = \sqrt{\beta(t)}$ ($\beta_{\min} = 0.01, \beta_{\max} = 20$)

**Misalignment**  We define the well-aligned region, $\Xi^*$, to be eight circles with radii equal to 0.4, and each circle centred on the circumference of the unit circle (such that the spacing between the circles is even). This geometry is inspired by the eight Gaussian dataset, commonly used for demonstrating generative models on simple datasets. For $p_{\text{ood}}$, we use the same form and parameters as the aforementioned $p_{\text{model}}$ but with $\alpha_1 = \alpha_2 = 0.5$. As such, in misaligned regions, the signal mean, $\mu(\theta, \xi)$ is systematically lower. We stress that in real-world scenarios, we have very limited knowledge of $p_{\text{ood}}$, and hence cannot write down an explicit $p_{\text{true}}(y|\theta, \xi)$.

**Design Strategies**  For an implicit design prior $p_{\text{ref}}$, a diffusion model trained to sample a Gaussian mixture model (GMM) with eight Gaussians with evenly-spaced means on the unit circle and diagonal covariance matrices ($\Sigma = \sigma^2 I_2; \sigma = 0.1$) could be used. Note that this design prior assigns high mass to regions in $\Xi^*$ and thus reflects our preferences towards areas of design space that exhibit lower misalignment. For a design prior that is a GMM, the time-dependent score $\nabla_{\xi_t} \log p_t(\xi_t)$ is available analytically [8]. As such, we simply use the analytic time-dependent score in the reverse process with `DiffBED`. For the Rank variants and Random, we also sample candidate designs from this GMM. BED, as in the main body of the paper, is unconstrained, gradient-based optimisation of designs. In this experiment, the observation $y$ is continuous. As such, we use the prior contrastive estimator (PCE) (Foster et al., 2020) for estimating the EIG and its gradients.

**Inference**  For all strategies, we perform posterior inference using a particle-based ULA scheme as described in Appendix C, omitting the projection step, with $N = 10^6$ particles.

**Evaluation**  We compute the incremental EIG of the chosen design at each iteration using the pCE estimator using $N = 9\,000, M = 110$ posterior samples (taken from the $10^6$ inference particles). The model exhibits non-identifiability with respect to the ordering of the two sources, $\theta_1$ and $\theta_2$ during posterior sampling. As such, the correspondence between estimated and true sources may be swapped between samples. To address this when computing the $L_2$, we evaluate both orderings of each posterior sample and use the ordering that yields the lower error. We average over 25 roll-outs with different random seeds and present our results in Figure 3.

In addition to the quantitative results presented in the main body (Figure 3), we present an aggregation of the experimental designs, across rollouts, for each design strategy in Figure 8.

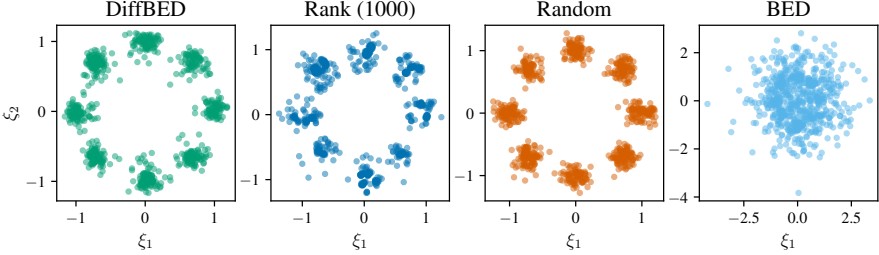

Figure 8: Designs aggregated across all iterations and experimental rollouts (25 seeds $\times$ 30 iterations) for 2D location finding. `DiffBED`, Random and Rank all leverage the implicit design prior in different ways to produce designs from regions where there is lower model misalignment. On the other hand, BED produces designs by *only* maximising the EIG, producing designs that lie in regions where there is greater misalignment.

---

[8]As our design prior is a GMM, the time-dependent score $\nabla_{\xi_t} \log p_t(\xi_t)$ under the variance-preserving diffusion process is available in closed form. Specifically, the forward process is $\xi_t = \sqrt{\alpha_t}\,\xi_0 + \sqrt{1 - \alpha_t}\,\varepsilon$, $\varepsilon \sim \mathcal{N}(0, I)$ and the marginal density at time $t$ is a Gaussian-convolution of the GMM, $p_t(\xi_t) = \sum_{k=1}^{8} \pi_k \mathcal{N}(\xi_t; \sqrt{\alpha_t}\mu_k, \Sigma + (1 - \alpha_t)I)$, which remains a mixture of Gaussians with inflated covariance. Hence the score has a trivial, analytic expression.

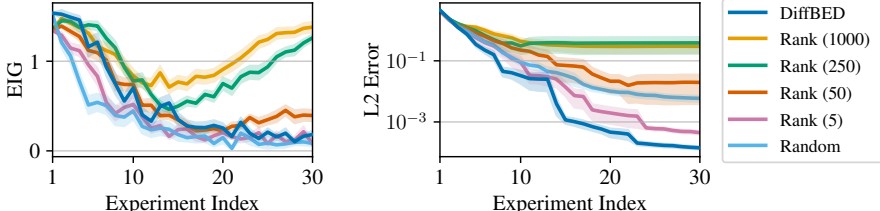

Figure 9: Incremental EIG (under $p_{\text{model}}$) at each experiment iteration (left) and average $L_2$ error between posterior samples and ground truth locations (right) for a larger range of size of candidate sets.

# E   ADDITIONAL IMAGE EXPERIMENTS

## E.1   CELEBA: ACTIVE PREFERENCE ELICITATION

In addition to the *search* experiments in the main body, we present an example of *active preference elicitation* on CelebA. The *search* experiments in the main body focus on recovering a single *observable* parameter $\theta$, often using designs that live in the same space as $\theta$ itself (e.g., the image search motivated by the criminal sketch problem). In contrast, *active preference elicitation* concerns learning a user's *latent* preferences, such as a set of preference weights over interpretable attributes or styles.

This problem class is widespread, with motivating examples ranging from recommender systems to personalization. For instance, a dating platform may wish to recommend profiles that a user is likely to engage with. A common modelling assumption is that a user's preferences can be summarized by weights over distinct, interpretable features (e.g., whether the image depicts glasses or a smile). Under this view, preferences can be inferred from pairwise comparisons, where the user indicates which of two candidate profiles they prefer.

Concretely, for this experiment, we take $\theta$ as a unit-normalised vector in which each element is a preference weighting for binary CelebA attributes. We use a Bradley-Terry (BT) response model, in which designs are pairs of images. We parametrise the latent reward as being proportional to the dot product between the preference vector and a vector of classifier probabilities that each attribute is present for each image in the pair. See Appendix A for additional experimental details and Appendix C for details about the inference.

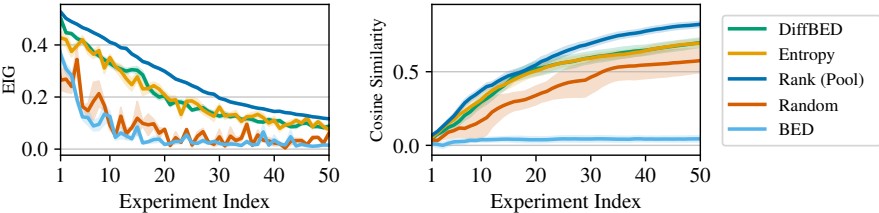

Figure 10: EIG and cosine similarity for preference elicitation on CelebA. Designs consist of image pairs with binary preference responses. Standard BED yields EIG values comparable to random selection and struggles to identify the true preference vector (low cosine similarity), while `DiffBED` maintains higher cosine similarity and more informative designs.

As shown in Figure 10, `DiffBED` achieves consistently high EIG designs and substantially outperforms both the standard BED and random baselines by producing much higher cosine similarities.

Interestingly, the EIG performance of the standard BED baseline is substantially lower than `DiffBED` and the other baselines. This suggests that gradient-based optimisation is struggling here, unlike in the search experiments, where BED produces consistently high-EIG designs, albeit totally unrealistic ones. A likely reason is that, in this setting, the EIG is defined through a classifier trained for discrete prediction, whose mapping from input to output is dominated by saturated confidence regions and sharp decision boundaries. In such regimes, gradients with respect to the input can be poorly conditioned, often near-zero or highly variable, making it difficult for gradient ascent to reliably improve the design objective.

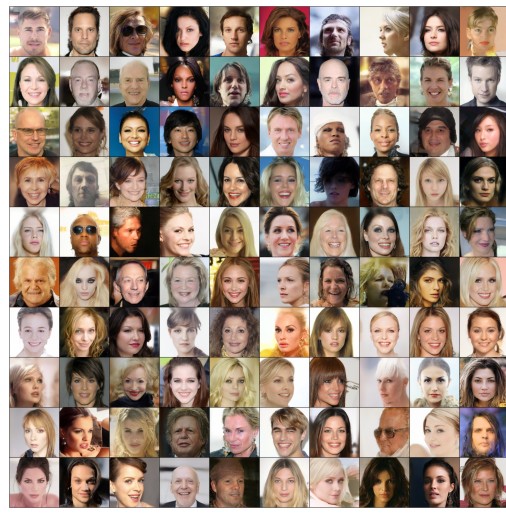 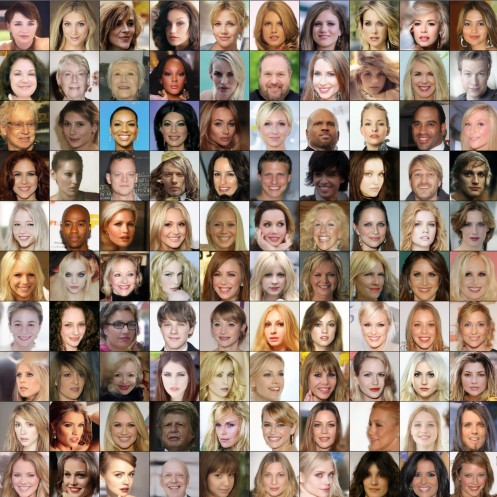

(a) `DiffBED` designs.                   (b) *Entropy* designs.

Figure 11: Example designs for the CelebA search problem.

By contrast, the reference prior in `DiffBED` helps guide optimisation toward sensible designs, assisting the method in locating regions of the design space with high EIG even when the EIG signal itself is weak or noisy. While `DiffBED` again outperforms *Entropy* on this problem, *Rank* slightly outperforms it. This likely relates to the poor performance of BED in this task, and to the quality of EIG gradients when the likelihood is parametrised by a discriminative classifier rather than a smoother, more continuous encoder. We thus find that *Rank* provides a useful variant of `DiffBED` for setups with more discrete features, but is less helpful when simple sampling from $p_{\text{ref}}(\xi)$ is insufficient to generate good candidate designs.

### E.2    CELEBA: INFORMATION-THEORETIC SEARCH

We present below additional experiments for information-theoretic search on CelebA.

**Example DiffBED Designs: CelebA**   Figure 11a shows 100 example images generated by `DiffBED` and Figure 11b shows the same for Entropy. This serves as a qualitative evaluation of our designs. Both are similarly high-quality images as they leverage the same underlying diffusion model, although with different objective functions for guidance. On the other hand, the designs produced by standard BED (Figure 12) are imperceptible from pure noise, despite their high EIG values.

**Ablations**   In Figure 13 we present results across the two extra values of the likelihood temperature parameter $\tau^{-1} = \{25, 50\}$, in addition to the value of $\tau^{-1} = 10$ used for CelebA in the main body of the paper. Larger values of $\tau^{-1}$ indicate less noise in the observation process, i.e., more informative outcomes, We see that larger $\tau^{-1}$ indeed leads to larger EIG values and a larger gap to the random baseline.

Here, we also present the performance of the naive BED baseline under data simulated from the assumed model which we call BED (assumed). That is, we sample $y \sim p(y \mid \theta, \xi)$ from the misspecified model likelihood, with no adjustments to the fact that the resulting designs may be meaningless. We see that BED (assumed) achieves both high EIG scores and cosine similarities in this case. However, the designs produced by BED in this case are imperceptible from pure noise (Figure 12) and could not be used in a real-world experiment. As soon as we sample data $y$ from a more realistic likelihood which returns uninformative data when the designs are pure noise, the cosine similarity of the naive approach (BED) drops to near zero, indicating a failure to determine the correct $\theta$.

In a second ablation, we study the effect of the number of the candidate designs considered in the *Rank* method. We plot the performance with $N = 100$, and $N = 10000$ candidate designs, alongside the $N = 1000$ set-up used in the main body of the paper. See Figure 14. Unsurprisingly, the

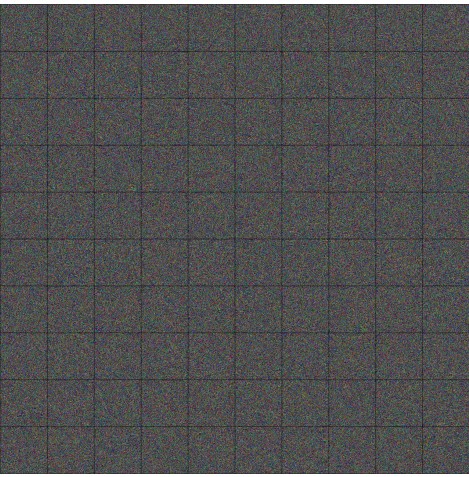

Figure 12: Example standard BED designs for the CelebA search problem.

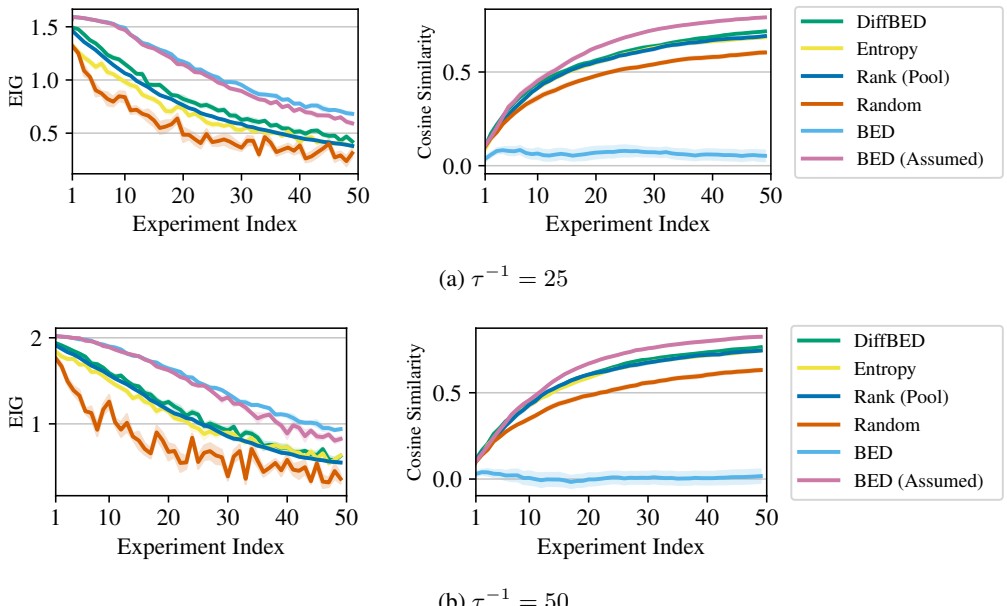

(a) $\tau^{-1} = 25$

(b) $\tau^{-1} = 50$

Figure 13: EIG and cosine similarity values for CelebA on the search problem, where outcomes are top-2 ranks of the images in the set of four design images. We now vary the value of $\tau$ used in the likelihood. Means are plotted $\pm$ one standard error.

performance of this baseline is a monotonic function of the pool size. However, increasing the pool size also comes with additional costs, especially when the pool is itself generated from a model, in which case $N = 10000$ would be significantly more time consuming than `DiffBED`.

### E.3 ZAPPOS

**Example DiffBED Designs: Zappos** We present 100 example designs produced by `DiffBED` and standard BED on the Zappos discrete rating problem. See Figure 15b and 15a.

**Ablations** We additionally present in Figure 16 results for $N = 10$ levels of discrete ratings, rather than $N = 5$ used in Section 6.3. Intuitively, we might expect that a more fine-grained rating would be more informative. Indeed, comparing against Figure 7, using $N = 10$ yields slightly higher cosine similarities.

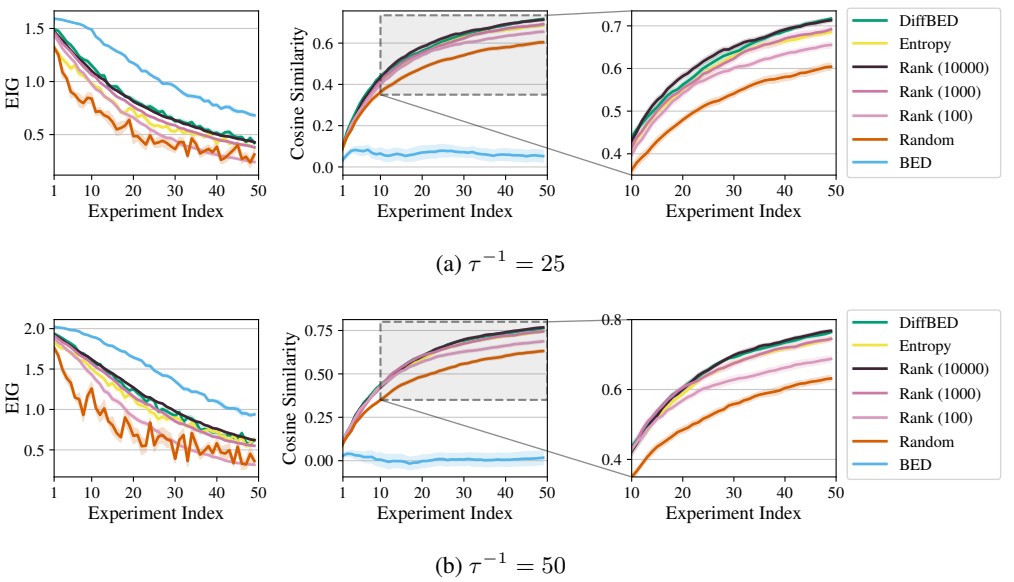

(a) $\tau^{-1} = 25$

(b) $\tau^{-1} = 50$

Figure 14: EIG and cosine similarities as we the number of candidate designs consider by the *Rank* baseline in the CelebA search problem. Means are plotted $\pm$ one standard error.

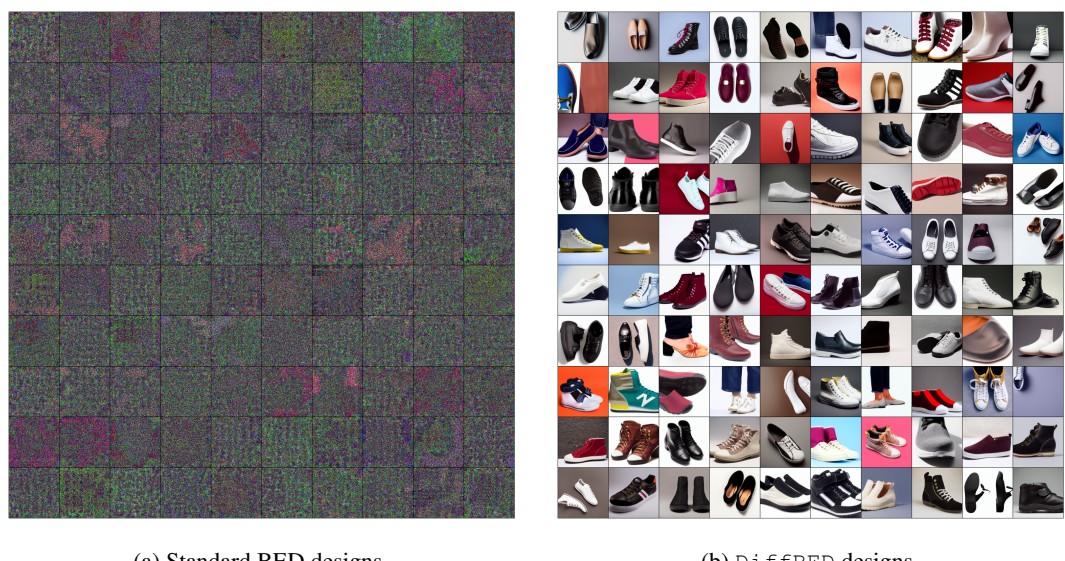

(a) Standard BED designs.

(b) `DiffBED` designs.

Figure 15: Example designs for the Zappos discrete rating problem.

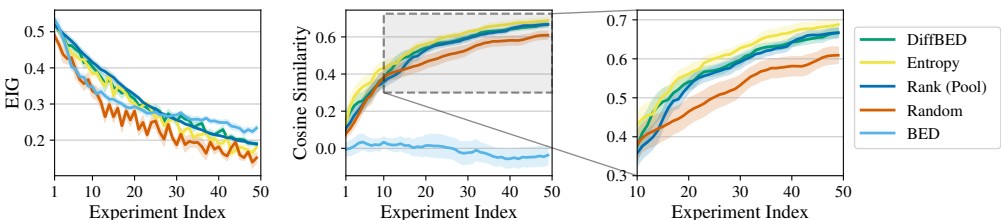

Figure 16: Quantitative results on the Zappos dataset, where designs are single images and outcomes are discrete ratings now on a scale from 1-10. Means are plotted $\pm$ one standard error.

