# OpenReview forum: "DiffBED: Scaling Bayesian Experimental Design to High-Dimensions"
_ICLR.cc/2026/Conference — ICLR 2026 Poster_

### Official Review · Reviewer_nkQr · 2025-10-31

**Soundness:** 3
**Presentation:** 2
**Contribution:** 2
**Rating:** 4
**Confidence:** 3

**Summary:**

This paper introduces DiffBED, a Bayesian experimental design method for high-dimensional design spaces that uses a pretrained diffusion model as a prior over feasible designs. The approach guides the reverse-diffusion process with gradients of expected information gain (EIG), producing proposals that stay on the data manifold (realistic) while remaining highly informative for learning the latent target. The authors argue that naïvely maximizing EIG in high dimensions can “reward-hack” likelihood misspecification; DiffBED mitigates this by sampling from an EIG-tilted prior rather than optimizing EIG directly. On image-level tasks—including 512×512 shoe designs—DiffBED outperforms standard BED and several strong baselines.

**Strengths:**

1. The paper clearly shows how naive EIG in high-dimensional settings can suffer from likelihood misspecification, which makes staying on the data manifold meaningful.

2. Exponential tilting of a diffusion prior is a clean way to sample realistic, informative designs—no extra training required.

3. This method uses a pretrained diffusion model, so there’s no retraining at every step which is time consuming for sequential BED.

4. This method works on large images (512×512) and outperforms standard BED.

5. This method improves the similarty to $\theta_{true}$ , which indicates true target recovery.

**Weaknesses:**

1. The paper doesn’t compare against a simple constrained EIG optimizer that stays on the data manifold (e.g., trust-region steps or directly maximizing EIG with a prior penalty). Without that apples-to-apples baseline, it’s hard to tell whether the gains come mainly from enforcing feasibility or from sampling the EIG-tilted prior itself.

2. Exponential tilting can collapse to a single mode on multimodal landscapes, and picking a good $\alpha$ is non-trivial because it trades diversity against feasibility.

3. The method leans on a strong pretrained diffusion prior. If an appropriate prior doesn’t exist or is domain-mismatched, it may be inapplicable, and tilting can’t add mass where the prior has zero mass.

4. Results focus on discrete outcomes; if y is continuous, EIG estimation and gradients are harder.

**Questions:**

1. Did you try a direct EIG optimizer that stays on the data manifold (e.g., trust-region steps or EIG + prior penalty)? What’s your intuition for how it would perform vs. DiffBED?

2. How sensitive are performance and diversity to $\alpha$?

3. On multi-peak landscapes for $p_{ref}(\theta)$, do you see mode collapse when $\alpha$ is small?

4. What happens when the prior is biased or domain-mismatched—does it still produce meaningful designs?

5. For continuous y, how would you estimate EIG, and which gradient estimator would you use (just a brief idea is fine)?

*** Duplicate word ***

line 222, by solving time-reversal of Equation (9)

---

> ### Author Response · Authors · 2025-11-22
>
> Thank you for your valuable and constructive feedback. We are encouraged to hear that you found DiffBED to be a practical and effective technique for high-dimensional BED, and that our demonstrations of the failures of traditional BED methods were clear.
>
> **1. Constrained EIG Optimizer**
>
> **W1 ("The paper doesn’t compare against a simple constrained EIG optimizer that stays on the data manifold (e.g., trust-region steps or directly maximizing EIG with a prior penalty). Without that apples-to-apples baseline, it’s hard to tell whether the gains come mainly from enforcing feasibility or from sampling the EIG-tilted prior itself.")**
>
> **Q1 ("Did you try a direct EIG optimizer that stays on the data manifold (e.g., trust-region steps or EIG + prior penalty)? What’s your intuition for how it would perform vs. DiffBED?")**
>
> We highlight that DiffBED is currently the **only method** for constrained EIG optimization on tasks with complex, high dimensional designs spaces and misspecified likelihoods.
>
> For problems with high dimensional designs (e.g., images), **the design prior is not explicitly known**, and instead **implicitly** defined through feasible designs. This rules out simple baselines like maximising the EIG plus a prior penalty or trust region methods, which would require **evaluating the likelihood of a design**. We stress that DiffBED can, in fact, be seen as a method for constrained EIG optimization. Key to our method is Equation (7), which trades off the EIG of a design for adherence to the design prior. In particular, if a design $\xi$ has zero mass under the reference prior $p^{ref}$, then the KL penalty will be infinite, and such a design will have zero mass under the tilted distribution. Further, if $\alpha \to 0$, we see that we will choose designs that maximize the EIG subject to the constraint that $p^{ref}(\xi) > 0$. Choosing larger values of $\alpha$ allows us to trade off smaller EIG values for higher likelihoods under the reference prior.
>
> **2. Continuous $y$.**
>
> **W4 ("Results focus on discrete outcomes; if $y$ is continuous, EIG estimation and gradients are harder.")**
>
> **Q5 ("For continuous $y$, how would you estimate EIG, and which gradient estimator would you use (just a brief idea is fine)?")**
>
> Our method does not require that $y$ is discrete, and can be readily applied when observations are continuous by using a standard EIG (gradient) estimator with DiffBED. Our experiments focus on discrete outcomes because a large number of real-world BED problems with high dimensional designs, for example with **humans-in-the-loop**, fall under this umbrella, and not because DiffBED is limited to discrete outcomes.
>
> To demonstrate this, we provide results on **source location finding** in our revised manuscript, a synthetic problem considered in the BED literature, as this problem has continuous observations. For this experiment, we used the prior contrastive estimator (PCE) as an EIG estimator [1]. However, we emphasise that our method is agnostic to the method of EIG gradient estimation, and any estimator of the EIG gradient is compatible with DiffBED.
>
> As with our core experiments, DiffBED samples **highly-informative points that adhere to the design prior**, reflected in its superior performance over Random and Rank baselines. Further, as before, traditional BED **catastrophically fails in inference and experimentation** in the model misaligned setting. Please see Appendix D for comprehensive experimental details and results.
>
> [1] A Unified Stochastic Gradient Approach to Designing Bayesian-Optimal Experiments. Foster et al., 2020.

---

> ### Author Response · Authors · 2025-11-22
>
> **3. Tilting / Choosing $\alpha$.**
>
> **Q2 ("How sensitive are performance and diversity to $\alpha$?")**
>
> **Q3 ("On multi-peak landscapes for $\alpha$  do you see mode collapse when $\alpha$ is small?")**
>
> **W2 ("Exponential tilting can collapse to a single mode on multimodal landscapes, and picking a good $\alpha$ is non-trivial because it trades diversity against feasibility.")**
>
> We would like to clarify that diversity and mode collapse are not issues for DiffBED. At each experiment iteration, we only wish to obtain a **single design** with high EIG while adhering to the design prior. Since we only produce a single design, we are not concerned with sampling many diverse designs at any given experiment iteration. At subsequent iterations, the design objective (EIG) itself has changed, meaning that we are not at risk of simply repeating designs across iterations. We demonstrate this in Figure 3.
>
> That being said, $\alpha$ is a hyper-parameter of our method which can be tuned. Higher values of $\alpha$ result in more informative designs, as per the assumed likelihood model, with lower adherence to the design prior. As such, choosing the highest value of $\alpha$ that still regularly produces realistic, implementable experiments is desirable. In our experiments, we conservatively set values of $\alpha$ that  produce designs, i.e. images, that are crisp and free of artefacts (Figure 8, and 13, present a wider set of designs from roll-outs). We expect that results of DiffBED can be marginally improved through a finer grid-search over parameter values, however, simple visual inspection of designs generated for a small set of $\alpha$ values allowed for effective values to be set.
>
> **4. Selecting a prior.**
>
> **W3 ("The method leans on a strong pre-trained diffusion prior. If an appropriate prior doesn’t exist or is domain-mismatched, it may be inapplicable…")**
>
> You are correct that a basic assumption of our work is the availability of a pre-trained diffusion prior over designs. However, without a design prior, we are in the traditional BED setting which can catastrophically fail to produce meaningful designs (Figure 1).
>
> We believe that having access to a diffusion design prior is a reasonable assumption:
> - DiffBED only requires a diffusion prior over designs. This is independent of the experimental outcomes, or $(y, \theta, \xi)$ parings. Thus, we are not at odds with our goal of collecting experimental data.
> - Diffusion models have become the de facto choice for learning generative priors, and strong pre-trained diffusion models are becoming increasingly available across a variety of domains.
>
> **W3 ("...tilting can’t add mass where the prior has zero mass.")**
>
> We agree. However, this is not our goal. We do not introduce tilting to *fix* the design prior, but rather to generate designs which adhere to the prior. We explicitly want to **avoid** regions where the prior has zero mass as these represent invalid or off-manifold designs.
>
> **Q4 ("What happens when the prior is biased or domain-mismatched—does it still produce meaningful designs?")**
>
> In scenarios with severe domain mismatch (e.g., shoes vs faces), we should not expect any method that enforces consistency with the design prior to producing meaningful designs.
>
> A more realistic scenario is where the design prior still produces meaningful designs, but may be an imperfect reflection of our preferences. In this case, DiffBED will still produce meaningful designs. We highlight that the hyper-parameter $\alpha$ allows us to control our reliance on the design prior, and a smaller value of $\alpha$ may be chosen to relax the prior constraint when the design prior is biased.

---

### Official Review · Reviewer_MFdK · 2025-11-01

**Soundness:** 2
**Presentation:** 3
**Contribution:** 3
**Rating:** 4
**Confidence:** 2

**Summary:**

This paper addresses the problem of scaling Bayesian Experimental Design (BED) to high-dimensional design spaces by first identifying a critical issue: directly optimizing Expected Information Gain (EIG) in high dimensions exploits model misspecification, producing unrealistic designs (e.g., noise images) where the likelihood is overconfident. The authors propose DiffBED, which constrains design optimization using a diffusion model as a reference prior over feasible designs, sampling from a distribution proportional to $p_{ref}(\epsilon) \cdot \exp(EIG(\epsilon)/\alpha). Designs are generated via information-guided diffusion, combining the diffusion model's score function with EIG gradients in the reverse SDE. The authors conducted experiments on image-based tasks, and DiffBED generates realistic and informative designs in high-dimension space, while standard gradient-based BED fails by producing noise despite high model EIG.

**Strengths:**

1. The problem studied in this paper is well-motivated, and the authors clearly justified that EIG optimization inherently seeks regions where the likelihood is overconfident, identifying model misspecification as the fundamental barrier to scaling BED.

2. The reward hacking analogy well motivates the approach, and by incorporating pre-trained diffusion models as reference priors, the proposed method prevents reward hacking while maintaining high informativeness.

3. Experimental results demonstrate that DiffBED is able to scale on high-dimensional settings.

**Weaknesses:**

1. The proposed method is highly dependent on having a high-quality, pre-trained diffusion model for the design space. This does not fully solve the high-dimensional problem as shift it from needing a perfect likelihood to needing a perfect generative prior. It would be great if the authors could elaborate more on this part.

2. The proposed method is computationally expensive as it requires running a full, guided reverse-diffusion process for each experimental iteration.

3. I think the authors should position the method as a hybrid, not a purely Bayesian one. It combines principled Bayesian inference (for $\theta$) with a pragmatic, regularized optimization (for $\xi$). The design objective $p^{*}(\xi)$ introduces $p^{ref}(\xi)$ as an external regularizer not derived from the original generative model.

**Questions:**

1. Please see the comments in Weakness part.

2. Could the authors provide more details on the BED baseline used in the experiments? I would suspect that the main computational bottleneck is the EIG gradients calculation, so how much does the proposed method cost more than the BED baseline?

3. It seems that the Rank baseline is highly competitive in the conducted experiments, can the authors discuss in more detail?

4. The proposed method depends on high-quality pre-trained diffusion models and encoders, which are feasible for vision but limited in other domains. Could the authors discuss the feasibility of applying DiffBED beyond vision (e.g., molecules, audio, text, tabular data), what happens when good diffusion models might not exist, and discuss when DiffBED is preferable over simpler pool-based approaches like Rank?

---

> ### Author Response · Authors · 2025-11-21
>
> Thank you for your review and positive comments on our motivation behind DiffBED and its scalability. We hope our responses below and our updated paper address the concerns you raised.
>
> 1. **Reliance on Diffusion**
>
> **W1 ("The proposed method is highly dependent on having a high-quality, pre-trained diffusion model for the design space. This does not fully solve the high-dimensional problem as shift it from needing a perfect likelihood to needing a perfect generative prior. It would be great if the authors could elaborate more on this part.")**
>
> We agree that DiffBED relies on access to a design prior. However, we stress that this prior does not need to be **perfect**. Its role is only to **regularize** the design optimization process. Using a less performant diffusion model in DiffBED may result in visually less appealing images, but crucially, images that still define **valid, implementable experiments**. In contrast, traditional BED requires a perfect likelihood, where even small model errors are exploited, leading to optimal designs that resemble **white noise**, and eventually catastrophic failure in experimentation and learning.
>
> Critically, learning a better likelihood ostensibly requires **more experimental outcomes**, i.e., a dataset of $(y, \theta, \xi)$ pairings. Such data is fundamentally scarce in the problems BED is meant to address. Further, the likelihood will always be grossly misaligned in regions where there exists no training data (e.g. gaussian noise design for a human image preference elicitation), once again leading to the hacking behaviour exhibited by traditional, unconstrained BED. In contrast, a design prior can be learned entirely from **unlabeled designs**, which we believe represents an important practical benefit.
>
> **First part of Q4 (” The proposed method depends on high-quality pre-trained diffusion models and encoders, which are feasible for vision but limited in other domains. Could the authors discuss the feasibility of applying DiffBED beyond vision (e.g., molecules, audio, text, tabular data), what happens when good diffusion models might not exist…”**
>
> DiffBED is readily applicable once we have access to a diffusion design prior and diffusion models have become the de facto choice for generative modelling, including beyond images. Strong pre-trained diffusion models are increasingly available, including in **video, audio, and molecules**. Further, in such domains, foundation models provide off-the-shelf access to powerful *specialised* design priors across tasks through conditioning e.g. text prompts.
>
> While it is true that DiffBED requires access to a generative diffusion model as a design prior, we note that **the use of encoders in our experiments is not a limitation of DiffBED itself**. Instead, it arises from the constraints of **explicit Bayesian modelling**, which requires access to a likelihood, $p(y|\theta, \xi)$. In the problem settings we study, no offline dataset of $(y, \theta, \xi)$ pairings exist from which one could directly learn this complex likelihood. This is, unfortunately, often the case in widely applicable problems, such as preference elicitation, as it is often impossible to record the true, data-generating $\theta$. Under these constraints, constructing a likelihood using an encoder, together with canonical parametric likelihood models (e.g., Bradley-Terry), is what **enables Bayesian inference, and traditional gradient-based BED methods** to be applied to these problems, not just DiffBED.
>
> **W2 ("The proposed method is computationally expensive as it requires running a full, guided reverse-diffusion process for each experimental iteration.")**
>
> In general, we find that DiffBED is sufficiently scalable for practical, high-dimensional BED uses. DiffBED requires **$\sim$30 seconds** to produce a high quality, high EIG design set (as seen in Figure 1) on a NVIDIA H100 GPU (250 reverse-diffusion steps). On the other hand, the standard BED baseline requires $\sim$ **3 seconds** (250 SGD steps). Note that using more, or **any amount** of, compute with the traditional BED baseline, for example through more SGD steps, does not resolve the fundamental issue of the reward hacking behaviour presented, wherein the likelihood function’s inaccuracies are exploited

---

> ### Author Response · Authors · 2025-11-22
>
> 2. **Not truly Bayesian**
>
> **W3 ("I think the authors should position the method as a hybrid, not a purely Bayesian one. It combines principled Bayesian inference with a pragmatic, regularized optimization")**
>
> We agree that this aspect of our work could have been explained more clearly. To clarify, BED is “Bayesian” in the sense of Bayesian decision theory [4], where we seek to maximize a utility function on designs $\xi \in \Xi$. This requires us to choose a space of feasible designs $\Xi$. Standard BED takes this set to be unconstrained, but we argue for choosing $\Xi$ to be the set of designs with $p^{ref}(\xi) > 0$. Maximizing the EIG over this set is still a Bayesian approach, even though we have added a constraint.
>
> However, since we do not have access to the likelihood $p^{ref}(\xi)$, we resort to sampling from $p^*(\xi)$, which can be loosely viewed as enforcing a soft constraint. We note that in the limit $\alpha \to 0$, [5] we do indeed recover constrained EIG optimization, which is Bayesian.
>
> [4] Bayesian Experimental Design: A Review. Chaloner et al., 1995.
>
> [5] Diffusion Models as Constrained Samplers for Optimization with Unknown Constraints. Kong et al., 2025.
>
>
> 3. **Baseline and Ablations Technical Clarifications**
>
> **Q2 ("Could the authors provide more details on the BED baseline used in the experiments? I would suspect that the main computational bottleneck is the EIG gradients calculation, so how much does the proposed method cost more than the BED baseline?")**
>
> The BED baseline performs direct, gradient-based optimisation of the design (sets).  Across all our experiments, we take 250 SGD steps. As the core image problems considered have a discrete observation space, we used the single nested EIG estimator for the gradients of the EIG [6, 7].
>
> Despite using a large number of particles in our inference scheme and EIG estimation ($N=10^{6}$), as the $\theta$ during experimentation lives in the latent space, a core innovation in our inference scheme, this is tractable. The BED baseline takes $\sim$ 3 seconds per design set of 4 images (Figure 1) on an NVIDIA H100 GPU. DiffBED is more expensive, requiring $\sim$ 30 seconds per design set in Figure 1, as it leverages a reverse diffusion process, which involves sequential calls to a diffusion model. As before, we stress that this larger cost is tractable, and that no amount of computation resolves the fundamental barriers of the standard BED approach, which exploits any level of inaccuracy in the likelihood model.
>
> [6] Automating Inference, Learning, and Design using Probabilistic Programming. Tom Rainforth. PhD thesis, University of Oxford, 2017.
>
> [7] Deep Bayesian Active Learning with Image Data. Yarin Gal, Riashat Islam, Zoubin Ghahramani, ICML 2017.
>
> **Q3 ("It seems that the Rank baseline is highly competitive in the conducted experiments, can the authors discuss in more detail?"),
> Q4 ("and discuss when DiffBED is preferable over simpler pool-based approaches like Rank?”)**
>
> We expect the gap between DiffBED and Rank to grow as the task becomes more difficult, as there may not be any designs in the pool which yield high EIG. DiffBED, in contrast, can directly generate high-EIG Designs, as it leverages **EIG gradient information**. For instance, we show DiffBED outperforms Rank when designs are sets of images (Figure 4), but can perform close to Rank when designs are single images (Figure 5).
>
> We further emphasize that Rank is not an existing baseline in the context of BED. This is an ablation of our own method, introduced for the first time in this paper to help demonstrate the contributions of different components in DiffBED.

---

### Official Review · Reviewer_tuUX · 2025-11-02

**Soundness:** 3
**Presentation:** 3
**Contribution:** 2
**Rating:** 4
**Confidence:** 3

**Summary:**

This paper presents an approach to apply diffusion models to propose informative and plausible designs for Bayesian adaptive experimental design in high-dimensional design spaces. The paper shows that, in such settings, likelihood model misspecification can severely misguide BED algorithms towards designs which have an apparent high expected information gain (EIG), though due to errors in the likelihood model, leading traditional methods to produce no longer meaningful designs in high dimensions. To mitigate that, the proposed method employs pre-trained diffusion models whose samples are guided by an EIG-based gradient estimator towards designs which are informative for the given task without the need to retrain the diffusion model. Experiments are presented on high-dimensional experimental design problems involving images.

**Strengths:**

* The paper is mostly well written and follows a clear structure which is relatively easy to follow.
* The proposed method seems relatively simple to implement within modern generative modelling frameworks.
* Some of the principles and insights in this paper may be applicable beyond its setting, such as the presentation on the issues with misspecification in BED, how a prior over feasible designs can help mitigate these issues, and how to apply pre-trained diffusion models to design tasks without a need for retraining.

**Weaknesses:**

* In Sec. 3, it is not clear how the decomposition of the EIG from Eq. 5 to 6 was derived. Eq. 5 was derived using the alternative formulation of the TEIG in terms of $\theta$, instead of $y$, which was the one presented in Eq. 4, making things confusing. Eq. 6 seems to have been derived by adding and subtracting TEIG from the model's EIG, but why the step in Eq. 5 was necessary is unclear.
* It is not explained how Eq. 11 was derived. I believe that might require some background in diffusion models, though such a derivation should at least be found somewhere in the appendix, given that the EIG guidance term is a crucial contribution from this paper.
* Experiments are presented on image design tasks, which escape traditional problems in experimental design, making it difficult to assess the impact of the paper outside this context. Despite the complexity of optimisation and sampling in high-dimensional design spaces, the probabilistic models in these experiments seem reasonably simpler than traditional models found in science and engineering applications where BED typically finds its applications. Hence, I'm unsure of this paper's impact.
* The presented performance plots show that, besides the gradient-descent BED baseline, the proposed DiffBED's performance is mostly very close to the performance of simpler baselines. Yet, I reckon that std. deviations, which I interpret as the shaded areas around the curves, are quite small as well. Therefore, the practical significance of this paper's contribution remains unclear.
* I'm also unsure about the claim in Sec. 5 that "the first to identify that model misalignment is not constant across design space and show the potential for reward hacking". Prior work in BED has discussed issues related to model misspecification and their effect on design optimisation, such as Foster et al. (2025), though to a different extent. In addition, the issue of model mismatch as a function of the designs is explicitly modelled in, e.g., Bayesian calibration frameworks by the model discrepancy/error term (Kennedy & O'Hagan, 2001), which have been recently applied to BED (Oliveira et al., 2024; Sürer et al., 2024). Additional references are listed below.

Minor:
* A few typos are present throughout the text, which require careful revision.
* Eq. 10 is missing a time differential $\mathrm{d}t$ next to the term in brackets.
* I believe it should be $\xi_t$, instead of $x_t$, in the inline equation on line 244.

References:
* Kennedy, M. C., & O’Hagan, A. (2001). Bayesian calibration of computer models. Journal of the Royal Statistical Society: Series B (Statistical Methodology), 63(3), 425–464.
* Oliveira, R., Sejdinovic, D., Howard, D., & Bonilla, E. V. (2024). Bayesian Adaptive Calibration and Optimal Design. 38th Conference on Neural Information Processing Systems (NeurIPS 2024).
* Sürer, Ö., Plumlee, M., & Wild, S. M. (2024). Sequential Bayesian experimental design for calibration of expensive simulation models. Technometrics, 66(2), 157-171.

**Questions:**

Please, see weakness points above. In addition, I have the following more specific questions.

* Have other high-dimensional experiment settings been considered, beyond images? As far as I understand, diffusion models could also be applied in lower-dimensional settings and compared against other BED baselines in more traditional problems. Some synthetic problems can have their dimensionality adjusted in ablation problems to show how the performance of standard methods potentially degrade as the dimensionality increases, while hopefully DiffBED maintains reasonable performance levels.

* Did each experimental trial have a different target image in Sec. 6.1? Or was the same image used across each experiment, with the different random seed only affecting optimisation behaviour?

---

> ### Author Response · Authors · 2025-11-21
>
> Thank you for your detailed review and feedback. We hope our responses below and our updated paper, which includes a synthetic BED experiment (Appendix D), address the concerns you raised.
>
> 1. **Images as our primary focus**
>
> **W3 (“Experiments are presented on image design tasks, which escape traditional problems in experimental design, making it difficult to assess the impact of the paper outside this context. … Hence, I'm unsure of this paper's impact.”)**
>
> We agree that our experiments depart from traditional BED settings, but this is intentional. Many critical data acquisition tasks that are naturally low-data and enable active experimentation, for example, learning the preferences of an individual, are typically overlooked by the BED literature because traditional methods break down and **not because these problems are unimportant**. DiffBED represents the first practical BED methodology for such high-dimensional designs, which we believe will have significant impact by expanding the scope of real-world problems BED can be applied to.
>
> **W3 (".. the probabilistic models in these experiments seem reasonably simpler than traditional models found in science and engineering applications where BED typically finds its applications")**
>
> We respectfully disagree that our likelihoods are simpler than those found in traditional applications. Several of our experiments are focused on modelling human preferences over **sets of images**, which is a complex, high-dimensional task. In particular, the likelihoods we employ use a learnt neural network encoder as a proxy for problem-specific, perceptual similarity. We refer to Appendix A.1 to highlight the complexity of the likelihood PMFs, particularly, the Plackett-Luce model over partial rankings.
>
> **Q1 (“... diffusion models could also be applied in lower-dimensional settings ... synthetic problems can have their dimensionality adjusted in ablation problems to show how the performance of standard methods potentially degrade as the dimensionality increases, while hopefully DiffBED maintains reasonable performance levels.”)**
>
> While we focus on images in our experiments, we emphasize that DiffBED is a general-purpose technique which is applicable on other domains. An important insight of our work is that **model misalignment** is a fundamental roadblock for high-dimensional BED, not necessarily the dimensionality itself. We agree that adding a synthetic BED experiment demonstrating this barrier would be beneficial. As such, we have run the **source location finding experiment**. We provide a complete set of experimental details and results in Appendix D, which we summarise below.
>
> During evaluation, we crucially introduce model misalignment that varies across design space, as this is a fundamental challenge in BED, and one that is addressed by the implicit design priors leveraged in DiffBED. Concretely, during rollouts, data is generated from a mixture distribution, which follows the assumed model if the design lies in certain regions of the design space and from another, *unknown* model otherwise.
>
> **The same trend from our core experiments holds**:  mean $L_2$ error (averaged over 25 seeds) after 30 iterations - **BED: $3.06$, Random:  $5.91\times10^{-3}$, DiffBED: $1.41\times10^{-4}$**.  Standard, gradient-based BED, which has no means for incorporating the implicit design prior, leads to pathological data collection and inference, which is worse than random design. Crucially, note that this occurs even in the low (two) dimensional design space considered in this problem. On the other hand, **DiffBED produces designs that (1) adhere to the design prior and are (2) informative** as it significantly outperforms BED and random design. Further, DiffBED, which runs diffusion on a single design, also outperforms Rank that considers 1000 randomly sampled designs even on this simple low-dimensional problem. Please refer to Figure 8 and 9 in our updated appendix for the $L2$ error and sequential prior contrastive estimator (a lower bound on the EIG) of designs across the rollouts, and an aggregated plot of the designs of different strategies.
>
> Our primary focus is on real-world, high-dimensional problems with learned likelihoods because in such settings, it is (1) infeasible to specify likelihood models that are accurate across the entire design space, and (2) any design priors that may exist are implicitly defined through feasible design samples (i.e. not available analytically). However, as demonstrated through our location finding experiment, DiffBED can also readily be applied in the synthetic problems traditionally studied in the BED literature.

---

> ### Author Response · Authors · 2025-11-21
>
> 2. **Comparison with Rank**
>
> **W4 (“The presented performance plots show that, besides the gradient-descent BED baseline, the proposed DiffBED's performance is mostly very close to the performance of simpler baselines. ... Therefore, the practical significance of this paper's contribution remains unclear.”)**
>
> We acknowledge that DiffBED can perform similarly to ‘Rank’ and ‘Entropy’ in some experiments. **However, ‘Rank’ and ‘Entropy’ are not existing baselines in the context of BED**. These are ablations of our own method, introduced for the first time in this paper to help demonstrate the contributions of different components in DiffBED.
>
> DiffBED consistently and clearly outperforms both random selection and traditional BED. Importantly, the only baseline that exists in the literature (standard BED without a design prior) fails dramatically in the high-dimensional, model-misaligned regime we are interested in. This failure is itself a novel and important empirical finding, as prior work has not evaluated BED in such real-world settings with learned likelihoods, and has not identified design priors as a solution.
>
> Altogether, given (1) the poor performance of existing baselines, (2) the strong performance of DiffBED and its ablations, and (3) the novelty of these ablations themselves, we believe our paper provides a meaningful and practically significant advance.
>
> 3. **Missed references**
>
> **W4 ("I'm also unsure about the claim in Sec. 5 that "the first to identify that model misalignment is not constant across design space and show the potential for reward hacking")**
>
> We sincerely thank the reviewer for these references! We acknowledge Oliveira, R. et al. (2024) and  Sürer, Ö. Et al. (2024) also consider when a model’s alignment depends on the design, and have added a discussion of these works to our paper on Lines 310-319.
>
> We still believe we are the first to show that ‘reward hacking’ style behavior occurs for gradient-based BED in complex high dimensional settings with learned likelihoods. In addition, the approaches in the aforementioned papers are not suitable for the problems we consider. In our setting, we cannot hope to learn an explicit design-dependent bias term, as this depends on access to experimental outcome data $(y, \theta, \xi)$, which is fundamentally scarce. However, we can instead define an *implicit* design prior through a generative model which assigns mass to areas of design space where we expect bias/misalignment to be low. Crucially, this only requires access to feasible designs, $\xi$.
>
> We believe Foster et. al (2025) is a typo, as we could not find this exact work, and that the review is raising Forster et. al (2025) [1] (please do clarify otherwise). In this case, while we briefly discuss [1], they focus on model misspecification as a binary, global property of a model, whereas our work focuses on another related but distinct issue of misalignment to the true model that depends on the design.
>
> [1] Improving Robustness to Model Misspecification in Bayesian Experimental Design. Alex Forster, Desi R Ivanova, and Tom Rainforth, AABI 2025.

---

> ### Author Response · Authors · 2025-11-21
>
> 4. **Technical clarifications**
>
> **W1 ("In Sec. 3, it is not clear how the decomposition of the EIG from Eq. 5 to 6 was derived...Eq. 6 seems to have been derived by adding and subtracting TEIG from the model's EIG, but why the step in Eq. 5 was necessary is unclear.")**
>
> Thank you for pointing out that the addition of Eq. 5 is confusing. You are correct that this step is not necessary for deriving our EIG-TEIG decomposition, and we have updated our manuscript with an updated derivation that avoids this step.
>
> **W2 ("It is not explained how Eq. 11 was derived. I believe that might require some background in diffusion models, though such a derivation should at least be found somewhere in the appendix, given that the EIG guidance term is a crucial contribution from this paper")**
>
> We agree that this step could have been explained more clearly. Equation (11) is based on recent advances in diffusion models [2, 3], and can be derived using techniques from stochastic optimal control. While the full derivation is nontrivial, we have added a discussion in Appendix B.4 with the full details.
>
> [2] Fine-Tuning of Continuous-Time Diffusion Models as Entropy-Regularized Control. Uehara et al., 2024.
>
> [3] Inference-Time Alignment in Diffusion Models with Reward-Guided Generation: Tutorial and Review. Uehara et al., 2025.
>
> **Q3 ("Did each experimental trial have a different target image in Sec. 6.1? Or was the same image used across each experiment, with the different random seed only affecting optimisation behaviour?")**
>
> Each experiment trial did indeed have a different, randomly selected target image.
>
> **Minor: We thank the reviewer for the catches! Said typos have been amended.**

---

### Author Response · Authors · 2025-12-02

### **Dear Area Chair**, ###

We thank you for your hard work under the unusual circumstances of this review cycle. Below, we summarise our paper, and provide a brief overview of the reviewer feedback and our rebuttals.

### **Firstly, we are encouraged that all reviewers recognise the two central and novel contributions of our work:** ###

1. We present DiffBED, the first method to scale BED to experimental problems with complex, high-dimensional design spaces, successfully performing gradient-based optimization in **750,000 dimensions**, orders of magnitude greater than prior work (See Figure 1; where designs are sets of 4 images of size 3 x 256 x 256).  [Reviewer MFdK: S3, nkQr: S2, 3, 4 & 5]

2. We motivate our method through a novel and clear demonstration that gradient-based optimisation of the EIG under learned likelihood models leads to reward-hacking. In doing so, we reveal a fundamental barrier preventing traditional BED methods from scaling to impactful settings where the complexity and dimensionality of the data-generating mechanism necessitate learned likelihoods (see Figures 1–2 and Section 3).  [Reviewer tuUX: S3, MFdK: S1 & 2, nkQr: S1]

### **In response to feedback, we have:** ###

1) provided results on additional BED experiment with a **continuous observation space**, namely source location-finding, reaffirming the wide scope of our method (**Appendix D**).

2) improved the clarity of the manuscript as follows,
   - **Lines 151-153:** Clarified mathematical derivation of the exploitation of model overconfidence
   - **Lines 311-319:** Added missed references on related works that discuss design-dependent model misalignment, and clarified/highlighted our innovations that enable BED to scale to high-dimensional problems
   - **Lines 409-415:** Added discussion of when and why DiffBED outperforms simpler ablations like Rank
   - **Appendix B.4:** Added background on sampling from tilted distributions using diffusion models


### **Below, we would like to further address points raised by reviewers:** ###

### **1) Scope of experimentation** ###

**a) Focus on discrete outcome experiments (Reviewer nkQr: W4 & Q5)**

A broad class of problems that require learned likelihoods to approximate the true, highly-complex data generating mechanism are those that have humans-in-the-loop. In such problems, experiments are generally set up to yield discrete feedback, for example, through the elicitation of ratings or rankings.

However, DiffBED is a general framework, compatible with any EIG estimator, and hence can naturally be applied to experiments with continuous observations. To showcase this, we have added results on source-location finding (see Appendix D). The key takeaways from our original experiments transfer: DiffBED produces designs that (1) lie in regions of space where misalignment is lower, and (2) are informative, outperforming standard BED, random design and Rank.

---

> ### Author Response · Authors · 2025-12-03
>
> **b) Focus on image problems**
>
> **Comparison to existing BED literature (Reviewer tuUX: W3, Q1):** We focus on images because they represent an important class of problems with applications in real-world systems, for example, product recommendation, preference elicitation, and efficient search. Our experiments tackling the aforementioned settings represent by far the highest dimensional design problems addressed by the BED literature.
>
> We hold that that our likelihoods on the image domain are by no means 'simpler probabilistic models' than those traditionally considered in the BED literature. Critically, all of the likelihood models we consider are functions of embeddings of high-dimensional designs, $\xi$, and/or $\theta$, obtained through neural-network based encoders. In comparison, the data-generating mechanisms typically considered in the literature are conveniently assumed to be sufficiently simple and low-dimensional so as to allow an analytical likelihood to be assumed. **However, we do agree that a traditional BED experiment would help cement our proposed approach among the BED community, and as such, present the source location finding experiment (see Appendix D).**
>
> **Extensions (Reviewer MFdK: Q4):** As we propose a general, diffusion-based framework, our method in fact does transfer to other design domains, such as molecules and audio. However, as was the case with images prior to our work, experimental problems in such domains are yet to be conceptualized, or tackled by the BED community. Advancing BED in these domains will require novel and clear formulations of (1) the quantities to be learned, $\theta$, (2) the experimental parameters, $\xi$, and (3) an effective Bayesian inference procedure, including reasonable definitions of the prior $p(\theta)$ and likelihood $p(y \mid \theta, \xi)$. As such, given our novel formulations of high-dimensional image problems, as well as extensive experimentation across different likelihoods, datasets, and diffusion priors of increasing scale, we leave further design domains for future work.
>
> ### **2) Reliance on a diffusion-based design prior** ###
>
> Access to a diffusion-based design prior is rightly a basic assumption of DiffBED. Without access to a design prior, we can only implement the standard BED baseline, which catastrophically fails when there is a learned likelihood. We refer the reader to Figure 1 for a demonstration. Further, given that we introduce a diffusion-based framework, off-the-shelf generative foundation models can readily be leveraged without the need for re-training. See Section 6.3 (Text-to-Image Foundation Models) for further details on how text-conditioning of a foundation model provides access to problem-specific design priors.
>
> **a) Access to a “perfect” design prior (Reviewer MFdK: W1)**
>
> We stress that the diffusion prior does not need to be perfect, as its role is to serve as a regulariser. Although a strong design prior is desirable, a less performant prior would result in lower quality designs, e.g. images, but crucially, designs that still define meaningful experiments. On the other hand, the traditional BED always fails to produce meaningful designs, as unconstrained optimization inevitably exploits any imperfections in the learned likelihood model.
>
> **b) Domain-mismatched design prior (Reviewer nkQr: Q4)**
>
> In settings where the design prior is catastrophically misaligned, any method enforcing adherence to a design prior will fail, e.g. applying a diffusion model trained over clothes to problems on shoes.
>
> However, for the much more realistic scenario of a slight mismatch, we refer to results in Section 6.3 (Text-to-Image Foundation Models), where the design prior (Stable Diffusion v1.5) used is pre-trained on a large corpora of data beyond Zappos. As such, the design prior not only assigns probability mass to the Zappos dataset, but also to other data seen during pre-training. However, DiffBED is still able to leverage the generative capabilities of the design prior, and produces much more meaningful designs than the traditional BED baseline.

---

> ### Author Response · Authors · 2025-12-03
>
> ### **3) Comparison to baselines** ###
>
> We reiterate that no existing methods in the literature scale to high-dimensional design optimisation with learned likelihoods. As such, in our experiments, direct gradient-based optimization of the EIG serves as the primary baseline (See BED vs DiffBED in Figure 1).
>
> **Existing approaches to modelling misalignment (Reviewer tuUX: W5):** In high-dimensional design spaces, the feasible design space is not available analytically and must therefore be learned. For design spaces as complex as images, unlike prior work [1, 2] we cannot hope to learn an explicit design-dependent bias term, as this would require access to large amounts of experimental outcome data $(y, \theta, \xi)$, which is fundamentally scarce. Instead, we learn an _implicit_ design prior that places mass on regions of the design space where we expect bias or misalignment to be low. Crucially, learning such a prior only requires access to feasible designs. We note that there exists no BED methods that leverage an implicit design prior to constrain optimisation. Thus, our contribution is not limited to the specific instantiation of our method, but also to the broader idea of incorporating a powerful implicit design prior within the BED framework.
>
> **Other potential approaches (Reviewer nkQr: W1, Q1):** We opt for the most performant class of generative models, i.e. diffusion-based models, as design priors to maximise the applicability of our work. We note that other, ostensibly simpler ways of maximising the EIG while adhering to a reference prior (rather than sampling from the tilted distribution), such as adding prior penalties, require evaluating the prior likelihood, which state-of-the-art generative models (e.g. diffusion/flow matching) do not reliably provide.
>
> **Performance of Rank (Reviewer MFdK: Q3, tuUX: W4):** Rank can be viewed as design optimization using a dirac-delta prior over samples, and is thus an ablation of our proposed approach. No other work has considered restricting design optimization using a design prior to mitigate reward hacking in learned likelihood models. As such, its relatively strong performance is not a weakness, but instead supports a core insight from our paper. We note that in experiments where the search space for designs is largest, DiffBED clearly outperforms Rank, including for producing design-sets of four images (see Figure 4). We have amended the manuscript with discussion on the relative performance between Rank and DiffBED on lines 409-415.
>
> We remain fully available to address any questions or details you may need to assist your decision. We hope that we have addressed the main points raised by reviewers, and we would be grateful if you would increase the scores as you see fit.
>
> Thank you,
> Authors
>
> References:
>
> [1] Oliveira, R., Sejdinovic, D., Howard, D., & Bonilla, E. V. (2024). Bayesian Adaptive Calibration and Optimal Design. 38th Conference on Neural Information Processing Systems (NeurIPS 2024).
>
> [2] Sürer, Ö., Plumlee, M., & Wild, S. M. (2024). Sequential Bayesian experimental design for calibration of expensive simulation models. Technometrics, 66(2), 157-171.

---

### Meta-Review · Area_Chair_1cdm · 2026-01-08

**Summary:**

The paper identifies a fundamental barrier in high-dimensional BED: model misspecification. The authors demonstrate that in high-dimensional spaces, standard optimization procedures inevitably lead to reward hacking, where the system exploits flaws in the likelihood model to produce pathological or noisy designs that are meaningless in real-world applications. To solve this, the authors introduce DiffBED, a framework that utilizes a diffusion model as a generative prior to ensure designs remain on a feasible manifold.

After careful investigation on the paper I find this topic interesting and the authors have presented the first method to scale BED to experimental problems with complex, high-dimensional design spaces.

**Reviewer Concerns:**

Reviewer tuUX noted that the transition between Equation 5 and Equation 6 (the EIG decomposition) was unclear and appeared to contain unnecessary steps. They also pointed out that Equation 11, the core "EIG guidance term," was presented without a derivation or background in diffusion theory.

Reviewer nkQr expressed concern that "exponential tilting" (the method used to guide the diffusion) might cause the model to collapse to a single mode, losing diversity in designs. They also noted that choosing the temperature parameter $\alpha$ is non-trivial, as it creates a difficult trade-off between design diversity and realism.

Several minor errors were flagged, such as missing time differentials in Equation 10 and incorrect variable subscripts in the text.2.

**Reviewer Scores:**

The author seem to have address all the concerns raised, It makes sense for me to assume that the reviewers would raise they scores if they had been able to participate fully in the discussion.

---

### Decision · Program_Chairs · 2026-01-26

Accept (Poster)